# Coinfection with chytrid genotypes drives divergent infection dynamics reflecting regional distribution patterns

Tamilie Carvalho [1,2,8 ✉], Daniel Medina [1,3,4,8], Luisa P. Ribeiro [1], David Rodriguez[5], Thomas S. Jenkinson[6], C. Guilherme Becker[4], Luís Felipe Toledo [1,8] & Jessica L. Hite[7]

By altering the abundance, diversity, and distribution of species—and their pathogens—globalization may inadvertently select for more virulent pathogens. In Brazil's Atlantic Forest, a hotspot of amphibian biodiversity, the global amphibian trade has facilitated the co-occurrence of previously isolated enzootic and panzootic lineages of the pathogenic amphibian-chytrid (*Batrachochytrium dendrobatidis*, 'Bd') and generated new virulent recombinant genotypes ('hybrids'). Epidemiological data indicate that amphibian declines are most severe in hybrid zones, suggesting that coinfections are causing more severe infections or selecting for higher virulence. We investigated how coinfections involving these genotypes shapes virulence and transmission. Overall, coinfection favored the more virulent and competitively superior panzootic genotype, despite dampening its transmission potential and *overall* virulence. However, for the least virulent and least competitive genotype, coinfection increased both *overall* virulence and transmission. Thus, by integrating experimental and epidemiological data, our results provide mechanistic insight into how globalization can select for, and propel, the emergence of introduced hypervirulent lineages, such as the globally distributed panzootic lineage of Bd.

[1] Laboratório de História Natural de Anfíbios Brasileiros (LaHNAB), Departamento de Biologia Animal, Instituto de Biologia, Unicamp, Campinas, São Paulo, Brazil. [2] Department of Ecology and Evolutionary Biology, University of Michigan, Ann Arbor, MI 48109, USA. [3] Sistema Nacional de Investigación, SENACYT, Building 205 City of Knowledge, Clayton, Panama, Republic of Panama. [4] Department of Biology, and Center for Infectious Disease Dynamics, One Health Microbiome Center, The Huck Institutes of the Life Sciences, The Pennsylvania State University, University Park, PA 16802, USA. [5] Department of Biology, Texas State University, San Marcos, TX 78666, USA. [6] Department of Biological Sciences, California State University—East Bay, Hayward, CA 94542, USA. [7] School of Veterinary Medicine, Department of Pathobiological Sciences, University of Wisconsin-Madison, Madison, WI 53706, USA. [8] These authors contributed equally: Tamilie Carvalho, Daniel Medina, Luís Felipe Toledo. ✉email: tamilie@umich.edu

Globalization can facilitate the emergence and spread of infectious pathogens[1,2]. When hosts are exposed to completely novel pathogen species or genotypes, coinfections with multiple genotypes can allow new pathogen variants to emerge via recombination between genotypes[3,4]. Genetically diverse or "mixed" infections can also select for more virulent pathogens owing to within-host competition for resources[5]. For instance, competitively superior genotypes generally exploit host resources faster, and thus, limit the reproduction and transmission of prudent (less virulent) pathogens[6]. Moreover, even if more aggressive pathogens kill their hosts and reduce their own fitness, less virulent pathogens suffer a disproportionate disadvantage (linked to reduced transmission) and may be eliminated by natural selection[5,7,8]. In addition, "overall virulence", defined as the virulence resulting from the interaction between all coinfecting genotypes and the host[5], can surpass or be below the virulence of the most and least virulent pathogens, respectively, or have an intermediate level[5]. However, empirical studies examining genetically diverse infections in wild, nonlaboratory host–pathogen systems, remain limited. A central research objective, therefore, is to understand how mixed-genotype infections affect disease outbreaks in the wild, which will enable the design of more effective public health and conservation interventions[3,4].

In Brazil's Atlantic Forest, a hotspot of amphibian biodiversity[9], globalization has led to the co-occurrence of enzootic (Bd-Asia-2/Brazil) and panzootic (Bd-GPL) lineages of the amphibian-killing fungus *Batrachochytrium dendrobatidis* (Bd), which have different introduction histories. In addition to these two lineages, novel recombinant genotypes (hereafter "hybrids") have also emerged[10–12]. Bd is a cutaneous fungus that disrupts key physiological functions of amphibian skin[13], leading to the potentially fatal disease chytridiomycosis[14]. Bd genotypes from the enzootic and panzootic lineages vary in phenotypic traits, competitive ability, and virulence[15–18]. In addition, hybrid genotypes, such as the ones resulting from the recombination between Bd-Asia-2/Brazil and Bd-GPL, can be more virulent relative to their parental lineages; as seen in one host species during an experimental trial using single-genotype infections[18].

Notably, in Brazil, most amphibian declines have occurred in hybrid zones; locations where Bd-Asia-2/Brazil, Bd-GPL, and hybrid genotypes co-occur[10,19]. This observation suggests that coinfections may either lead to more severe infections or select for more virulent lineages, which can lead to the replacement of less competitive enzootic genotypes in the pathogen population[17]. The co-occurrence of enzootic, panzootic and hybrid genotypes of Bd in Brazil's Atlantic Forest suggests that coinfections among these genotypes carry important implications for disease dynamics and virulence evolution[11,20]. Therefore, it is crucial to understand whether coinfections could shape competitive interactions among these genotypes and alter pathogen transmission and virulence (e.g., ref. [17]).

Here, we examined how coinfection between four divergent genotypes of Bd from two distinct lineages and a hybrid affect competitive interactions within hosts. Then, we asked whether these interactions altered pathogen virulence (host survival and life span) and transmission potential (using Bd load as a proxy for future transmission potential). Coinfection selected the most virulent genotype (P1), despite decreasing its transmission potential and *overall* virulence. However, *overall* virulence increased in coinfections involving the least competitive and least virulent genotype (P2). These experimental results, combined with our field data, provide mechanistic insight into how within-host competitive interactions could lead to the distribution pattern of Bd lineages across Brazil's Atlantic Forest.

## Results

First, we examined the potential confounding effect of the initial inoculum dose on host life span (a metric of *overall* virulence), competitiveness (early establishment), and transmission potential (total pathogen load). Overall, any changes in single versus coinfection treatments were found to depend on the underlying pathogen genotype. The initial pathogen dose contributed little to virulence (host life span), competitiveness, or transmission potential (all terms $P > 0.05$). See results of this analysis in Supplementary Material; Supplementary Fig. 1. Thus, it is unlikely that the observed outcome of coinfection is a by-product associated with a reduced inoculation dose of the reference Bd genotypes in these coinfection treatments.

To examine how coinfection affects disease outcomes, we used genotypes from the Global Panzootic Lineage (Bd-GPL; "P1" and "P2"), the enzootic lineage Bd-Asia-2/Brazil ("E1" and "E2") and their hybrid ("H") (Table 1 and Fig. 1A, B); and performed analyses for each genotype alone or when in mixed infections with one or two other genotypes. We considered the genotypes from the most recently derived Bd-GPL as the reference genotypes, which we placed in competition with the two enzootic and hybrid genotypes in an effort to capture the arrival of Bd-GPL in Brazil—where coinfections with the local enzootic lineage led to the emergence of the hybrid. Our results show a link between genotype virulence and competitive ability, but the direction of this relationship varied across particular Bd genotypes and their introduction history (i.e., enzootic vs. panzootic). The most competitive genotype, the panzootic P1, was also the most virulent one (i.e., causing the lowest average life span of the host), whereas the enzootic and hybrid genotypes were less competitive than the panzootic genotypes and showed intermediate virulence. During the trial, frogs from the control group did not become infected or experience mortality, suggesting that mortality was directly associated with experimental pathogen infections.

We determined that the panzootic genotype P1 was the fastest to establish and outgrow all other genotypes in single-genotype infections (Bd load on day 21 of P1 relative to all other genotypes: $P < 0.001$; Fig. 2A). However, under coinfection scenarios, P1 suffered competitive suppression. In all coinfections involving P1, early establishment was suppressed relative to the single-infection trials (GLM single/mixed: $P < 0.001$; Fig. 2B). For instance, the degree of competitive suppression of P1 was strong, particularly

## Table 1 *Batrachochytrium dendrobatidis* (Bd) genotypes used in the challenge assay.

| Isolate | Designation | Lineages or genotype | Locality | Host species | Year | Passage |
|---|---|---|---|---|---|---|
| CLFT 168 | P1 | Bd-GPL | Alto Caparaó, MG | *Phantasmarana apuana* | 2015 | 7 |
| CLFT 198 | P2 | Bd-GPL | Santa Isabel, SP | *Aquarana catesbeiana* | 2016 | 4 |
| CLFT 041 | E1 | Bd-Asia-2/Brazil | Morretes, PR | *Bokermannohyla hylax* | 2013 | 15 |
| CLFT 172 | E2 | Bd-Asia-2/Brazil | Pindamonhangaba, SP | *Aquarana catesbeiana* | 2016 | 5 |
| CLFT 024-02 | H | Hybrid | Morretes, PR | *Hylodes cardosoi* | 2011 | 20 |

The table includes isolate/genotype name, designation (panzootic "P", enzootic "E" and hybrid "H"), chytrid lineages or genotype, locality (municipality, state), host species, year of isolation, and the number of passages for each isolate.

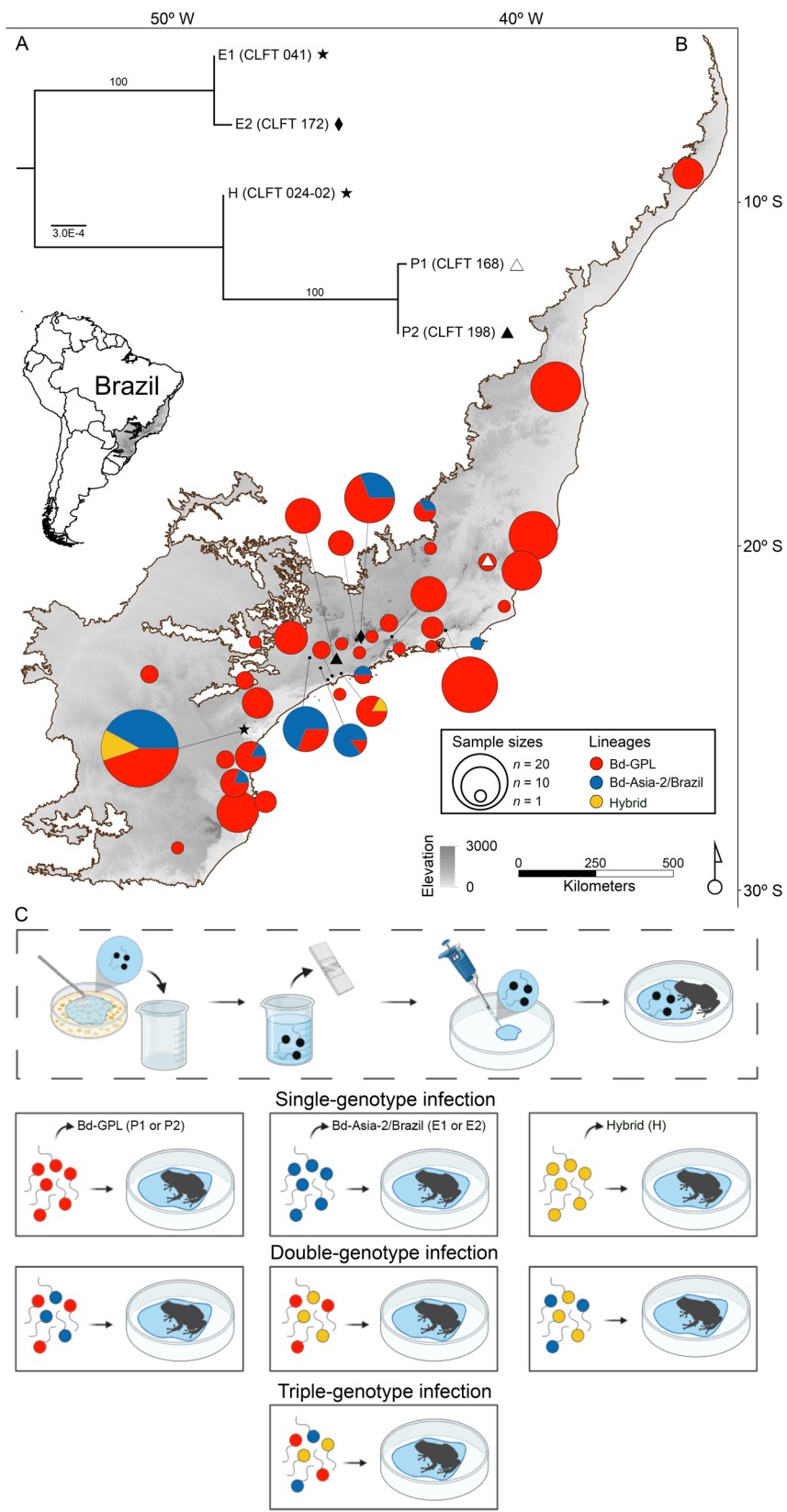

in triple infectiosn involving E1 and double infections involving E2 and H, but relatively weak in triple infections with E2 (Fig. 2B). These results underscore that virulence evolution may depend strongly on genotype-specific traits.

Competitive suppression also reduced the *overall* virulence and transmission potential of P1. Specifically, single infections with P1 resulted in the greatest reduction in host life span (GLM single/mixed: $P = 0.052$; Fig. 3B) and survival (log-rank test: $X^2 = 12.50$, df = 5, $P = 0.03$; Fig. 3E). In most cases, host survival rates slightly increased when coinfected (that is, virulence decreased). This trend was evident in the two-genotype infection with the hybrid genotype, although this trend was not statistically

**Fig. 1 Phylogenetic relationships among *Batrachochytrium dendrobatidis* (Bd) genotypes, geographical distribution of their lineages, and experimental design for single- and mixed-genotype infections. A** Phylogenetic tree of Bd genotypes used in the challenge assay. **B** Spatial distribution of Bd lineages or hybrid genotypes identified from multiple amphibian species across the Brazilian Atlantic Forest (highlighted area in map) (Supplementary Data 2). On the map, symbols represent the location where we isolated the pathogen genotypes used in this experiment. White triangle: P1, black triangle: P2, diamond: E2, star: E1 and H. **C** Infection assay procedure. In order to prepare the Bd solutions, we flooded Bd cultures with distilled water, quantified Bd zoospore density using a hemocytometer, and adjusted all solutions to the same concentration (see "Methods" for additional experimental details). **C** was created with BioRender.com.

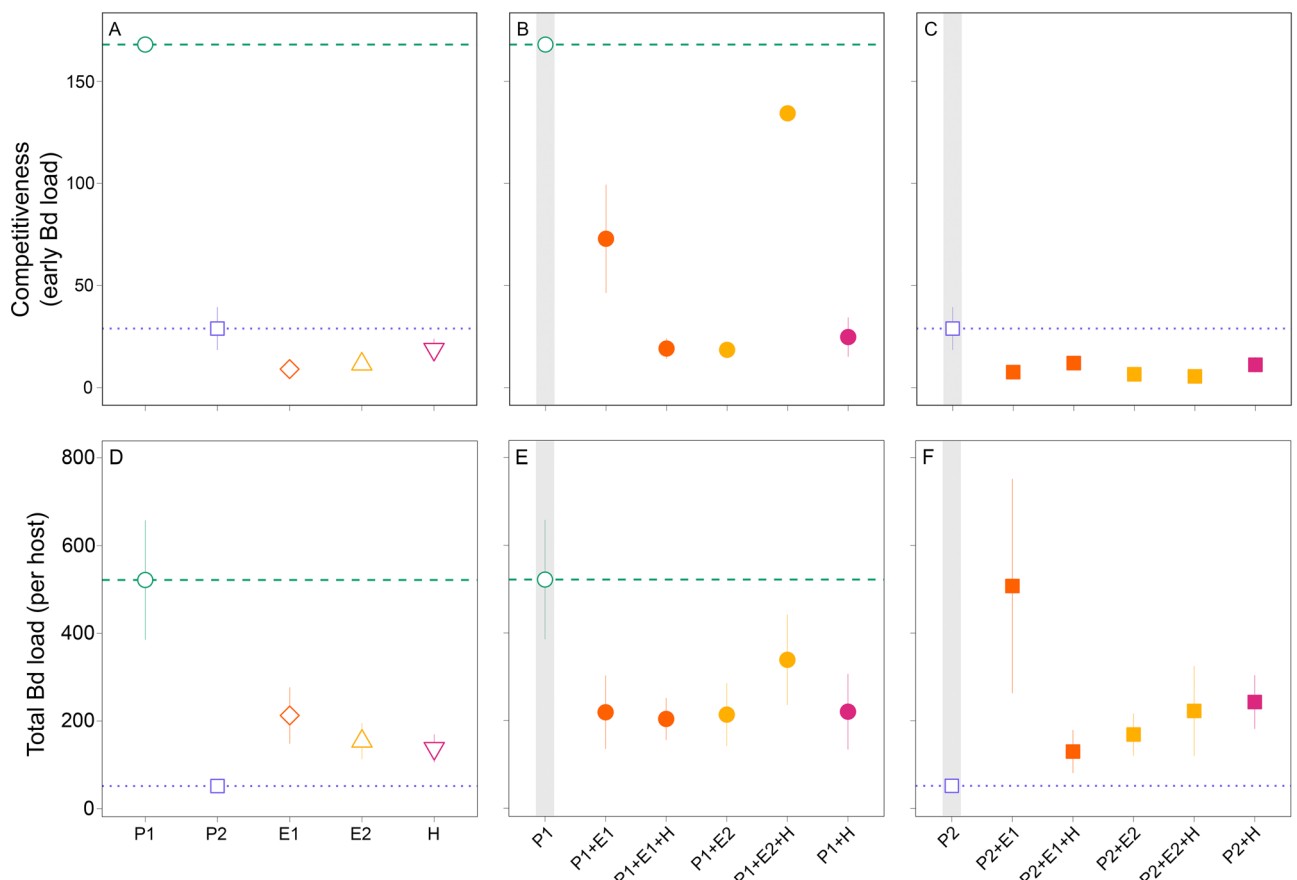

**Fig. 2 Competitiveness and transmission potential of *Batrachochytrium dendrobatidis* (Bd) in both single- and mixed-genotype infections. A–C** The competitiveness of each genotype is represented as mean Bd load (± standard deviation [SD]) during the first time point of the assay (day 21 post inoculation). **D–F** Transmission potential is represented as the average Bd load (± SD) after death or on day 76 at the end of the experiment. Competitive suppression could either help reduce the transmission potential of more virulent genotypes (P1; **E**) or increase the transmission potential of less virulent genotypes (P2; **F**). Gray shades in figures highlight single-genotype infections. Infection loads are represented by Bd concentration in pg/µl.

significant (median survival time P1: 31.5 days; P1 + H: 48.5 days; adjusted $P = 0.09$). These longer life spans, however, did not always lead to an overall increase in transmission potential. In all coinfections involving P1, transmission potential (total Bd load—after death or on day 76) was lower than in single-genotype infections (GLM single/mixed: $P = 0.025$; Fig. 2E), despite coinfections being mainly comprised by P1 (Fig. 4). Thus, even though coinfected hosts lived longer, coinfections with the enzootic genotypes and a hybrid still suppressed transmission potential for the most virulent genotype (P1) relative to the single-genotype infection.

In contrast, P2 was slower relative to P1 to establish and replicate in single infections and therefore, had a low competitive ability (Fig. 2A). P2 also showed similar pathogen loads at early stages of infection to all other infection treatments ($P > 0.05$; Fig. 2C). In mixed infections with P2, it is likely that facilitation occurred for two reasons. First, the total Bd load (transmission potential) was higher in coinfections relative to single-genotype

infections (GLM single/mixed: $P < 0.001$; Fig. 2F). For E1 in particular, coinfections with P2 lead to drastic increases in total Bd loads with levels reaching those similar to P1 single-genotype infections (Fig. 2D vs. F). Second, the proportion of P2 in these coinfections was substantially lower relative to either the E1 or E2 genotypes (below 0.25, Fig. 4A). In triple infections, however, this facilitation was hindered by the hybrid genotype (H), which competitively suppressed the more virulent genotype but enhanced the less virulent genotype (Fig. 4A vs. C). Importantly, despite the increase in pathogen loads (Fig. 2F), coinfections involving P2 did not show substantial increases in *overall* virulence (Fig. 3C, F). For instance, coinfections with P2 had no effect on mean life span (GLM single/mixed: $P = 0.113$; Fig. 3C) but slightly decreased host survival rates (log-rank test: $X^2 = 13.30$, df = 5, $P = 0.02$; Fig. 3F).

As for the Bd isolates collected from native species in the wild and *A. catesbeiana* individuals from frog farms, we determined that Bd-GPL is more broadly distributed in the sampled region

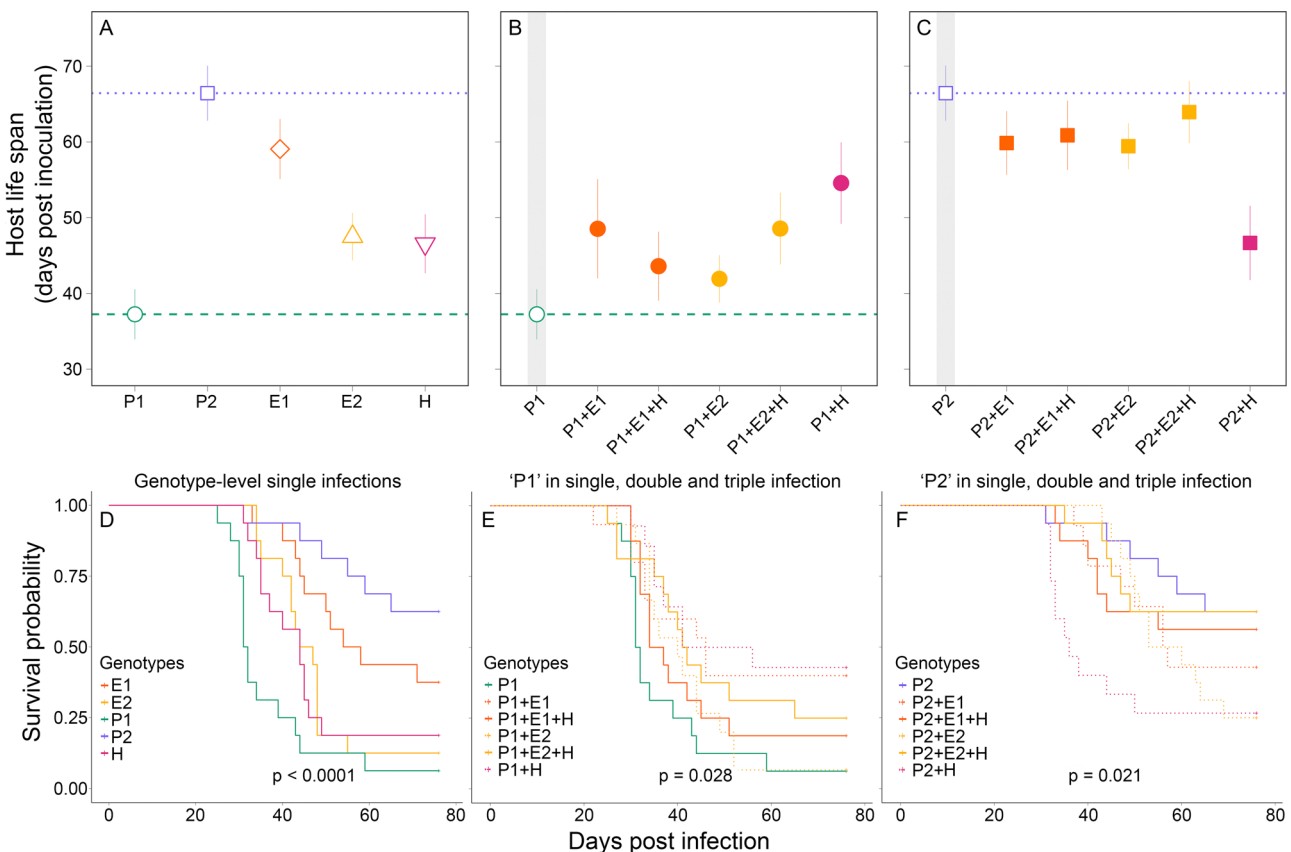

**Fig. 3 Variation in virulence across infection treatments.** Virulence, measured as host life span (**A**–**C**) and survival probability (**D**–**F**), varied across hosts infected with single- and mixed-genotypes of *Batrachochytrium dendrobatidis* (Bd). The presence of multiple genotypes reduced the *overall* virulence (i.e., increased host life span and survival) in those treatments involving the most virulent genotype (P1; **B**, **E**). Coinfections with the least virulent genotype (P2), however, did not influence host life span (**C**), though they had an effect on survival or median survival time (**F**). For survival curves, *P* values are from log-rank tests. Gray shades in figures highlight single-genotype infections. Points represent the mean host life span and bars the standard deviation.

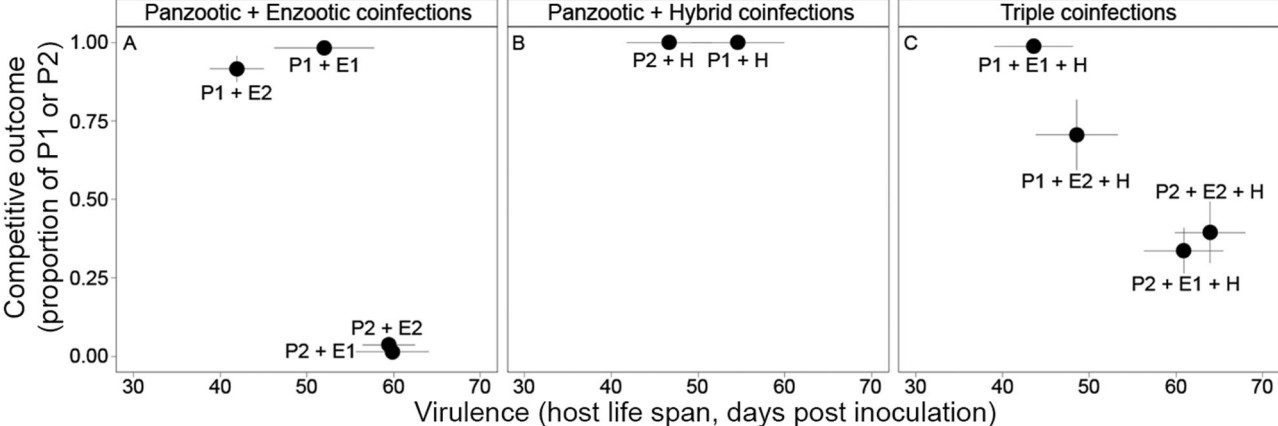

**Fig. 4 Competitive outcomes in mixed-genotype infections of *Batrachochytrium dendrobatidis* (Bd). A**–**C** Competitive outcomes for the mixed-genotype infections were based on the proportion of the total pathogen load (in pg/µl) that was either the most and least virulent genotype (P1 and P2, respectively).

(with 80%, $n = 190$, of the isolates belonging to this lineage), compared to the Bd-Asia-2/Brazil lineage (17%, $n = 41$) and hybrid genotypes (2%, $n = 5$) (Fig. 1B).

## Discussion
Coinfections by multiple pathogen genotypes are common in humans (reviewed in ref. [21]), plants[22,23], and wildlife[24,25]. Theory predicts that coinfections have the potential to fundamentally

alter infectious processes within and between hosts, and thus, they have important implications in epidemiological dynamics and the evolution of virulence (reviewed in ref. [5]). We show that coinfection can have differential outcomes on virulence and transmission potential over the course of infection depending on the combination of pathogen genotypes and their introduction histories. For a highly competitive and virulent genotype, such as P1 from the Bd-GPL lineage, these within-host interactions among pathogen genotypes could have led to both global[11] and

regional patterns of Bd distribution (e.g., Fig. 1B) and outbreaks. We found that some genotypes are competitively suppressed during coinfections, while other genotypes benefit from coinfections. For the most virulent and competitive genotype (P1), competitive suppression during coinfection reduced both *overall* virulence and transmission potential (total Bd load). The decline in transmission potential was likely driven through competitive suppression such that even though coinfected hosts lived longer, which could lead to higher shedding rates, overall pathogen production was reduced relative to hosts infected with a single pathogen genotype. For less virulent genotypes (e.g., P2 and E1), however, facilitative interactions among pathogen genotypes had little effect on *overall* virulence but did increase transmission potential. Thus, while highly virulent genotypes may be subject to competitive suppression during coinfections, less competitive and virulent genotypes may facilitate other genotypes leading to increased transmission potential.

Links between competitiveness and virulence in the genotype P1 are consistent with global epidemiological patterns of this pathogen. For instance, Bd-GPL is both globally distributed and associated with mass mortalities of amphibian hosts[26,27]. Our experimental results align with a previous study that demonstrated that Bd-GPL was more competitive than the enzootic lineage in the first four weeks post inoculation[17], a critical phase for successful pathogen colonization. Importantly, the outcome of the within-host competitive interactions observed in our study, and in ref. [17], may be captured by the observed distribution of Bd lineages, where Bd-GPL showed a much broader distribution compared to the enzootic lineage and the hybrid (Fig. 1B). Combined, these results suggest the likely scenario that upon arrival Bd-GPL rapidly spread throughout the Atlantic Forest outcompeting the enzootic lineage[10,17], which may be persisting up to this day, in part, due to within-host facilitative interactions.

In contrast to P1, which was isolated from a wild host, the less competitive and less virulent genotype P2 was isolated from a bullfrog farm. Such controlled environments with high density and low diversity of hosts can favor the emergence of highly virulent genotypes by increasing transmission potential[28]. However, the lower competitive ability, virulence and transmission potential observed in P2 may, in contrast, indicate that conditions associated with farms can also select for less virulent genotypes, possibly due to a constant availability of hosts. In addition, a high and constant availability of hosts may also allow pathogens to, for example, reallocate resources to suppress the host immune system. Thus, the evolution of pathogen virulence under farm conditions may be complex and difficult to predict, as suggested by Ribeiro et al.[29], where distinct Bd genotypes with different degrees of virulence were isolated from bullfrog farms.

The pathogen genotypes examined here span a range of relatedness and introduction histories (i.e., enzootic vs. panzootic). Theory predicts that the competitive outcomes among coinfecting pathogens depends on their genetic relatedness[28,30–32]. Distantly related pathogens are predicted to compete more strongly than closely related (and cooperating) pathogens. However, we found that both facilitation and strong competition occurred among the most distantly related genotypes, P2 + E1 vs. P1 + E1, respectively. Fully testing current theoretical predictions in this system would require a larger and logistically challenging experiment with both intra- and inter-lineage combinations that was beyond the scope of this current study. In the meantime, these results underscore an important point that is often overlooked by current theory: genotype by genotype interactions and introduction history may play stronger roles than relatedness per se.

The consistency of virulence patterns in Bd-GPL from both field observations and empirical studies spanning multiple host species is somewhat surprising. Variation within Bd-GPL in

phenotypic traits (e.g., zoosporangia and zoospore size) have been observed in laboratory experiments[16,17] and field studies[15], linked to pathogen incidence in natural populations[15] and virulence in experiments[33,34]. In addition, over multiple generations, competitive suppression could select for P1 to invest greater resources into phenotypically plastic transmission stages such that coinfections could actually lead to higher relative transmission[35]. Within this context, pronounced phenotypic variation in genotypes, such as P1 (and other Bd-GPL genotypes), may therefore help explain its ability to successfully outcompete other genotypes, establish a successful infection within-individual hosts and spread through the population, regardless of the high level of virulence. This observed link between competitiveness and virulence in P1 is consistent with theory[8,35] and empirical studies in several systems[6,23,36–39].

Our results agree with previous studies and help to explain why pathogens in general, and Bd in particular, can be highly virulent to their hosts despite their reliance on hosts resources for their own growth and fitness. An important effort for future studies is to understand the extent to which greater phenotypic plasticity in Bd-GPL may enable it to maintain fitness in novel environments and reduce the costs of virulence that are predicted to limit transmission. Identifying the particular genes or pathways that enable such plasticity may offer a novel target for managing outbreaks in Bd and other fungal pathogens.

One potential limitation of coinfection experiments, including ours, is that it can be difficult to tease apart the effects of coinfection per se and initial pathogen dose, which is often linked with virulence. We applied the same inoculation dose in both single- and coinfection treatments, which is common in coinfection studies using genotypes of one pathogen species[23,38,40,41], though not when competing two pathogen species (e.g., refs. [42,43]). This experimental design enabled us to measure competitive interactions between multiple pathogen genotypes while holding dose constant. That is, we asked, all else equal, how does coinfection influence pathogen growth, transmission potential, and *overall* virulence? Within this context, we showed opposite patterns in the infection dynamics of coinfections with P1 and P2, while also finding a non-significant effect of inoculation dose, suggesting that there is a stronger effect of genotype composition and their potential interactions.

An alternative question that is also addressed in coinfection studies is how does a competitive advantage/disadvantage (based on initial dose) influence competitive interaction, transmission potential, and virulence. Many of these studies manipulate coinfection with a twofold increase in pathogen dose. In this case, coinfection is still confounded by potential density-dependent effects or potential dose-dependent effects on virulence and transmission potential. interactions. These concerns can be limited by a careful selection of experimental doses such that a twofold increase in the experimental dose has a negligible effect on pathogen dynamics, virulence, or final pathogen load[6,36]. For the Bd system, however, such links between an increase in initial pathogen dose and virulence remains unclear[44]. Hence, we chose the more conservative experimental design to hold total pathogen dose constant.

Viewed in this light, neither study design can tease apart the effect of dose on coinfection and both designs are limited by the possibility that density-dependent effects could drive outcomes either by increasing harm to hosts through a dose response or by suppressing/releasing pathogen growth due to density-dependence. An important gap in knowledge is whether these two experimental designs reveal important differences from transient dynamics early in infection versus equilibrium levels after an infection has established[45]. Nonetheless, both designs address relevant and interesting questions. However, addressing

each of these questions in a single study is logistically challenging, especially when considering the sample sizes required to address this question with the number of pathogen genotypes examined here.

Moreover, while neither of these conditions likely mirror infections outside of the laboratory, they are experimentally tractable and commonly used (see Supplementary Table 1); ideally, one would examine coinfections in a response-surface design with the pathogen serovars or genotypes manipulated across a wide range of doses. Unfortunately, for most in vivo systems, including the focal system here, the latter design is logistically challenging and rarely examined. In reviewing over 20 coinfection studies, only one study manipulated dose across single- and mixed infections[46]. Our study, however, takes an important step in addressing one component of the coinfection puzzle and highlights important areas for future research.

Also, amphibians are recognized as an ideal system to assess the ecological and evolutionary implications of coinfections to both the infectious agent and the host[47], as they are commonly infected by multiple pathogens (e.g., refs. [48–51]) and have experienced disease-induced population declines globally[14]. Though, elucidating the factors shaping within-host interactions and scales (spatial and/or biological) at which coinfections influence disease dynamics in amphibians has proven to be highly complex and inconsistent, which has hampered our ability to identify general trends thus far (reviewed in ref. [47]). Previous work assessing coinfections by amphibian pathogens associated with population declines (e.g., Bd, *Batrachochytrium salamandrivorans* and *Ranavirus*) have determined that density-dependent competition among pathogens[52], order of pathogen arrival[53,54] and timing of exposure between pathogens[53,55] play key roles in parasite colonization and transmission potential, and influence amphibians' response and survival. Hence, our study adds to our current understanding by providing evidence of within-host competitive and facilitative interactions between genotypes of Bd, which can influence disease dynamics, virulence evolution, and shape the genetic structure of the pathogen population.

Understanding how these within-host dynamics play out in the wild over multiple host and pathogen generations carries important implications for the evolutionary trajectory of Bd and any strategies aimed at virulence management in an increasingly globalized society. Together, this work underscores the need for ongoing Bd monitoring efforts that include tracking coinfections of multiple Bd genotypes.

## Methods
### Infection assay
*Pathogen genotypes and amphibian host.* The Bd isolates used in our infection assays were obtained from tadpoles of captive North-American bullfrogs, *A. catesbeiana* (P2 and E2), and wild anurans (P1, E1 and H) (Table 1).

Prior to the experiment, isolates were maintained at 4 °C and passaged every four months. The number of passages ranged from 4 to 20 (Table 1). Although virulence attenuation can occur in Bd isolates serially passaged for long periods[56,57], Bd isolates with as much as 50 passages (compared to isolates with 10 passages) can present higher rates of zoospore production and overall population growth, and induced higher infection loads and prevalence in inoculation experiments[58]. Thus, we infer that the number of passages did not, at least to a noticeable level, affect probability of Bd colonization and reproduction in our assay.

To verify the phylogenetic relationships between our isolates in the challenge assays, we inferred a tree using published sequence data for these Bd genotypes (Supplementary Data 1). Using six MLST marker sequences (8009×2, BdC24, BdSC4.16, R6046, BsSC6.15, BdSC8.10) previously generated from associated studies[10,29], we constructed a hetequal distance dendrogram[10,59] using PAUP* version 4.0b10 (Fig. 1A). The support values for the major clades (Bd-GPL and Bd-Asia-2/Brazil) were estimated by bootstrapping over 100 replicates.

We used the terrestrial frog *Eleutherodactylus johnstonei* (Anura; Eleutherodactylidae), which is a direct-developing species (i.e., lacking a larval stage and with embryo hatching as froglets), native to the Lesser Antilles[60], that was introduced to the city of São Paulo, Brazil in 1995[61]. Frogs used in the experiment were collected from this introduced population. *E. johnstonei* is known to tolerate Bd infections and is considered a potential reservoir species in Dominica and Montserrat[62]. However, in Brazil, the introduced population of *E. johnstonei* have remained restricted to a small urban habitat and free of Bd. Two recent studies, after testing 100 frogs in total, did not detect Bd at this location[63,64]. Besides, this host population exhibited high levels of susceptibility to Bd in a previous experiment[64], therefore serving as a good model species for our experiments.

At the collection site, we selected adults with approximately the same size (SVL mean = 21.12 ± sd 2.66 mm, $n = 95$), and handled each adult frog with a new pair of disposable gloves to avoid potential Bd cross-contamination, and gently placed them into individual containers. We then transported the frogs to the laboratory in refrigerated coolers. To confirm that all hosts were Bd-free prior to experimental inoculations, we swabbed the skin of all the frogs upon arrival at the laboratory following a standard protocol[65]. We stored swabs at −20 °C until processing.

After swabbing, we housed the frogs individually in plastic enclosures (22 × 15 × 8 cm), with a layer of sphagnum moss that was previously autoclaved and moistened with distilled water. Frogs were housed in temperature-controlled rooms at 20 °C on a 12-h day–night cycle. We monitored frogs twice a day throughout the experiment and fed them calcium-enriched crickets ad libitum twice a week.

*Experimental exposure.* Experimental treatments included single-genotype exposures and mixed-genotype exposures with either two or three genotypes with all possible inter-lineage combinations (Fig. 1C). In total, the experiment included 15 Bd exposed treatments and one control group for a total of 265 hosts of the invasive species *E. johnstonei* (Supplementary Table 2).

To prepare Bd suspensions for the challenge assay, we transferred liquid cultures of each genotype to individual agar plates containing 1% tryptone and allowed them to grow for eight days at 11 °C under dark conditions. We then collected zoospores of each Bd genotype by flooding culture plates with 2 ml of distilled water for 45 min to induce zoospore release and then gently scraping the surface to maximize collection. From each zoospore suspension, we prepared standard inoculation solutions (concentration $2.98 \times 10^6$ zoospores/ml) by collecting a 1 ml subsample and quantifying the zoospore density using a hemocytometer (Fig. 1C). We did not filter the zoospore suspensions as zoosporangia were not observed on the hemocytometer.

For pathogen exposure, individual hosts were placed in Petri dishes with 1.5 ml of zoospore suspension containing a fixed inoculum dose ($2.98 \times 10^6$ zoospores/ml), and were exposed for 45 min at 20 °C (Fig. 1C). Frogs from the control group were exposed to 1.5 ml of distilled water. Individuals from coinfection treatments (those exposed to two or three genotypes) were exposed to a zoospore suspension containing equal proportions of each Bd genotype (with a final volume and inoculum dose of 1.5 ml and $2.98 \times 10^6$ zoospores/ml, respectively). By using such Bd dose, we aimed to reduce frog's mortality, while maintaining

infection, to increase our ability to track within-hosts pathogen dynamics throughout the experiment. This experimental design enabled us to address the following question. All else equal, how do coinfections influence pathogen growth, transmission potential, and *overall* virulence to the host? Importantly, the link between pathogen dose and virulence is still unclear in the Bd-amphibian system as evidenced by the large overlap between experimental infection studies that did or did not find a dose-dependent effect on survival (see Fig. 7 in ref. [44]).

To quantify Bd infection following exposure, we collected skin swabs on day 21 post exposure and after death or at the end of the experiment (day 76) following a standard protocol[65]. Both collection days span multiple Bd life cycles[66]. Swabs were extracted using the DNeasy blood and tissue kit (Qiagen, Valencia, CA, USA) following the manufacturer's protocol, with a minor modification that consisted of an extended incubation time (four hours) in the lysis step to increase DNA yield.

*Animal use ethics.* This study was approved by the Unicamp Animal Care and Use Committee (CEUA #5398-1/2019), Instituto Chico Mendes de Conservação da Biodiversidade (SISBio #71780-1), and Sistema Nacional de Gestão do Patrimônio Genético e do Conhecimento Tradicional (SISGen #A8246D0).

*Genotyping of coinfections.* To simultaneously detect Bd lineages (and hybrid) and estimate infection loads during experimental coinfections, we developed a genotyping assay for this study. Overall, we leveraged two genotyping assays to detect the presence or absence of Bd-Asia-2/Brazil, Bd-GPL, and the hybrid[67] in a mixed sample. Assay Bdmt_26360 targets a mitochondrial SNP and distinguishes between Bd-Asia-2/Brazil (allele A, 6-FAM) and Bd-GPL or hybrid (allele G, VIC)[17]. When allele G was detected in a mixed sample by the mtDNA assay, then we performed a second genotyping reaction using assay BdSC9_621917_AC (present study; Supplementary Table 3) to target a SNP in the nuclear genome to determine whether the hybrid genotype (allele C, 6-FAM) was present or absent. Absence of allele C would indicate only Bd-GPL was present in the sample (Supplementary Fig. 2). This nuclear assay allowed us to accurately distinguish between Bd-GPL and hybrid in 100% of the frogs from the double infection and over 80% from the triple infection treatments.

We used the skin swab DNA extractions as input for qPCRs in 20 µl volumes composed of 6 µl of the sample template, 10 µl of TaqMan Fast Advanced 2X Master Mix (Applied Biosystems, Inc.), 3 µl of nuclease-free water, and 1 µl of the SNP assay mix (Applied Biosystems, Inc.), at 20× concentration (18 µM forward primer, 18 µM reverse primer, 4 µM 6-FAM probe, and 4 µM VIC probe). On a QuantStudio 3 (Applied Biosystems, Inc.), cycling conditions consisted of 95 °C for 20 s, then 95 °C for 1 s and 60 °C for 20 s (data collection step) for 50 cycles. We analyzed amplification curves for either VIC or 6-FAM using the Standard Curve application that is part of the Thermo Fisher Connect cloud-based software.

*Bd quantification using standard curves.* We sourced three genotypes from the Collection of Zoosporic Eufungi at the University of Michigan to serve as genotyping positive controls and quantification standards. These included previously characterized isolates CLFT 041 (Bd-Asia-2/Brazil), CLFT 095 (Bd-GPL), and CLFT 024-02 (hybrid), of which two (CLFT 041 and CLFT 024-02) were used in the infection assay. Each isolate was transferred to 10 ml of 1% Tryptone liquid media and grown at 17 °C for 14 days in cell culture flasks. After confirming the presence of active zoospores, the medium was agitated and 1 ml was transferred to 1.5-ml centrifuge tubes that were spun at 10,000 rcf for

10 min to pellet the zoospores. The liquid was decanted off and the pellets were used as input for the GenJet DNA extraction kit (Thermofisher Inc.) following the manufacturer's recommendations but using 100 µl final elution volumes.

Replicate tubes were pooled and the resulting purified DNA was quantified on a Qubit 3 (Applied Biosystems). DNA for each isolate was normalized to 1.34 ng/µl. Serial dilutions with a dynamic range of $1.34 \times 10^{-1}$ to $1.34 \times 10^{-5}$ ng/µl were used as inputs for qPCR standards for each genotype and assay. Then, genotyping amplification curves were used to estimate Bd load measured in ng/µl based on the genotype-specific standard slopes (Supplementary Table 4; see ref. [68]). For statistical analyses, we used Bd measures (represented by quantified DNA concentrations) from the mitochondrial assay (Bdmt_26360), and transformed them to pg/µl to facilitate the calculation of parameter estimates and interpretation of analyses.

*Statistical analyses.* To ensure that our infection methodology was consistent across treatments, we first quantified the proportion of hosts that became infected over the course of the experiment using Generalized Linear Models (GLMs) with binomial errors (the proportion of hosts that became infected did not differ across genotypes or coinfections; Supplementary Fig. 3A–C; Binomial GLM, all $P$ values > 0.05). Then, we tested for differences in competitive ability (using early establishment of Bd as a metric—21 days post inoculation), pathogen virulence (life span and survival), and transmission potential (using total Bd load—after death or on day 76) of each genotype when alone and when in competition with either one or two other genotypes using a combination of generalized linear models (GLMs), planned a priori orthogonal contrasts, and survival analyses.

For GLMs, we chose the most appropriate distribution for each model using the R package *fitdistrplus*. For virulence and transmission potential, we included only the animals that were infected (and excluded animals that were exposed but uninfected; $n = 3$: E1 + H = 1, P2 + H = 1, P2 + E2 + H = 1). To understand how coinfections affected key traits of the reference genotypes, we used planned a priori orthogonal contrasts[69–71].

We computed survival curves using the Kaplan–Meier method, which was implemented using the *survfit* function from the package *survival*[72], and compared survival curves by conducting log-rank tests using the function *survdiff* (also from the package survival). We conducted post hoc comparisons of survival curves using the function *pairwise_survdiff* from the package *survminer*[73], which computes adjusted $P$ values to correct for false discovery rate (we used the method "BH"[74]). We examined the effect of coinfections by comparing survival curves of Bd genotypes from the panzootic lineage in single-genotype infections (P1 or P2) with their respective mixed-infection treatments, which included the Bd genotypes from the enzootic lineage (E1 and E2) and the hybrid (H).

Given the logistical constraints on how pathogen load was manipulated in our experimental design (see "Discussion" for expanded explanation), we examined the effect of initial zoospore dose on host life span (a metric of *overall* virulence), competitiveness (early establishment), and transmission potential (total pathogen load). We first explored the relationship between initial dose and coinfection treatments across pathogen genotypes and then when accounting for genotypic-specific differences, where coinfection treatment and pathogen genotype were included as fixed effects. We were unable to include interactions due to limited sample size. Based on the distribution of response variables, we analyzed the data using generalized linear models with gamma error distributions and log link functions to account for skewness and heteroskedasticity. All statistical analyses were conducted in R statistical software version 1.4.11[75].

**Collection and compilation of field data of Bd isolates**. To examine whether observed within-host competitive outcomes in the experiment reflect distribution patterns of Bd lineages across Brazil's Atlantic Forest, we genotyped Bd isolates obtained from tadpoles of natives and introduced (i.e., *A. catesbeiana*) frog species sampled at multiple locations (Fig. 1B and Supplementary Data 2). The sampling of tadpoles from native species took place in natural ponds and streams, and the sampling of *A. catesbeiana* tadpoles took place in farms where they are bred for commercial purposes. We determined tadpole infection status by examining the degree of dekeratinization in the oral disc[76], and used this approach to select tadpoles for Bd isolation. We used the genotyping results from this survey as a preliminary attempt to link the results from the exposure experiment to field data. However, our ability to qualitatively make such a link may be limited by the potential variation in depigmentation rates across Bd genotypes/ lineages; and also, by the inconsistent relationship between infection status and degree of dekeratinization of tadpole mouthparts in some species from the Atlantic Forest[77]. In addition, we complemented our field data by including previously published results of Bd genotyping from adults and tadpoles sampled in the study region (Supplementary Data 2). In total, this dataset included 237 isolates (182 tadpoles and 55 adults) collected from 42 species between 1987 and 2016 (Supplementary Data 2).

The isolation, culturing, DNA extraction, and genotyping procedures used in all Bd isolates included in this study (experiment and field collection) follow those described in ref. [10].

**Reporting summary**. Further information on research design is available in the Nature Portfolio Reporting Summary linked to this article.

## Data availability
The data used for all statistical tests are available at Dryad (https://doi.org/10.5061/dryad. cc2fqz693).

## Code availability
The full code for all statistical tests is available at dryad (https://doi.org/10.5061/dryad. cc2fqz693).

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

## Acknowledgements

We thank Diego Moura-Campos, Janaína de Andrade Serrano, Joice Ruggeri Gomes, Mariana Retuci Pontes, Victor Fávaro Augusto, Carolina Lambertini, and Kerry Gendreau for technical assistance throughout the experiment. We also thank Skylar Hopkins for comments on the manuscript. Grants and fellowships were provided by São Paulo Research Foundation (FAPESP #2016/25358-3; #2018/08650-8; #2018/23622-0; #2019/18335-5), the National Council for Scientific and Technological Development (CNPq #300896/2016-6; #302834/2020-6), the Coordination for the Improvement of Higher Education Personnel (CAPES—Finance Code 001), the National Science Foundation (IOS #2303908, DEB #2227340), and startup funds from Texas State University to D.R. and from UW-Madison to J.L.H.

## Author contributions

T.C. conceptualized the study with contributions from CGB and LFT. T.C., L.P.R., D.M., D.R., and T.J. conducted the laboratory methods. T.C., D.M., and J.L.H. analyzed the data and drafted the manuscript. All authors carried out the study and critically revised the manuscript.

## Competing interests

The authors declare no competing interests.
