## [Peer Review File · Communications Biology]

Reviewers' comments:

Reviewer #1 (Remarks to the Author):

The authors integrate experimental and epidemiological data to provide mechanistic insights into how the within-host competitive interactions could lead to the distribution pattern of Bd lineages across Brazil's Atlantic Forest. They demonstrate that coinfection with chytrid genotypes drives divergent infection dynamics reflecting broad epidemiological patterns.

This is a very interesting study. The study is well performed and the results are clearly described. This paper will be of interest to a broad audience.

I only have some minor comments:

Material and methods section

Line 173: could the authors provide more details on the inoculum preparation? By scraping the surface, did they collect sporangia? Did they wash the spores before using them?

Line 180: did the inocula for the coinfection experiments contain 2.98×10^6 zoospores for each strain or 2.98×10^6 zoospores in total?

Reviewer #2 (Remarks to the Author):

The Brazilian Atlantic Forest *Batrachochytrium dendrobatidis* system is one of very few well-documented interlineage contact zones for this globally distributed pathogen. It's wonderful to see more work done on this system, even more so to see this attempt to link up genotype distribution patterns to mechanisms of infection. I am somewhat concerned about components of the study and hope the authors can address my points.

I have reservations regarding the genotyping strategy, how it bears on the detection of recombination in general and how it can be applied to discriminate amongst isolates in some of the experimental treatments. First, can the authors comment on the reliability of the stepwise genotyping strategy in reliably identifying recombinants? From my reading, the key assumptions are unidirectional transfer of Bd-Asia-2/Brazil genomes into Bd-GPLs (recombinants only carry GPL mtDNA) and that recombinants can be reliably distinguished from GPLs on the basis of a very limited number of marker loci. I've referred back to O'Hanlon 2018, and the WGS phylogeny indicates that both Brazilian GPLs and Bd-Asia-2/Brazil genomes should carry variation indicative of within-lineage recombination, or the introduction of multiple variants of each lineage into Brazil (eg, Bd-Asia-2/Brazil genomes are not monophyletic within the Bd-Asia-2/Brazil lineage). Either process is likely to include variation that is difficult to identify with simple marker systems. In any case, identifying recombinants using simple marker systems will miss evidence of recombination involving other regions of the genome. For the experiment, I'm assuming that the missing 80% of frogs in the 3 isolate treatments were heterozygotes for the second step of the qPCR? Doesn't the shortcoming of the stepwise genotyping assay mean that it is impossible to clearly discriminate between any mixed GPL and Bd-Asia-2/Brazil infection versus a pure hybrid? In other words, for all of the four 3 isolate treatments, 2 of 6 possible outcomes cannot be discriminated amongst? If this is the case, why was this not discussed in the methods and impacts on analyses clarified? It's a red flag for me when your results and discussion involves an argument of competitive suppression of a GPL and in 2 of the 6 treatments involving that isolate you may be underestimating the number of competitive outcomes that unambiguously favour this lineage.

My greatest concern regarding the experiment is that dose is confounded with genotype treatment in the design. I understand why you wanted to make the overall dose strength consistent across treatments, but in so doing you've reduced the concentration of any one isolate by half in the 2 isolate

treatments and by 2/3 in the 3 isolate treatments. This means that any effect of treatment on changes in infection outcomes cannot be clearly attributed to competition. The reduction in P1 infectivity in mixed isolate treatments could be driven by the fact these treatments had a reduced dose of P1. There's lots of data out there supporting the relationship between increased virulence and increased dose strength.

The link between the experiment and the spatial data set is not really investigated. There have been a few studies of the spatial distribution of Bd genotypes in Brazil, but it's unclear to me how the experimental design is set up to address spatial patterns. As far as I am aware, arguments have been made regarding environmental envelopes and host specificity. For example, Greenspan et al (2018) showed how comparative patterns of virulence were reversed between GPL and recombinants in different native host species. The title of the paper includes 'reflecting broad epidemiological patterns', but I don't see these patterns investigated at all here.

The study boils down to a single experiment, and in that experiment, possible outcomes of 4 of 14 treatments cannot be identified, and the fundamental premise that outcomes are driven by competition cannot be fully tested due to confounding effects. I would be really happy if the authors can change my perceptions and look forward to seeing the revisions that will do so.

Reviewer #3 (Remarks to the Author):

The work of Carvalho et al. investigates how coinfections with 3 different Bd strains (BdGPL, BdBrazil and its hybrid) shaped virulence and transmission. They used both experimental and epidemiological data to elucidate the implications for disease dynamics and virulence evolution.

The manuscript is really well written and the obtained results are interesting and certainly deserve publication. The performed experiment is well planned and executed, and it really brings valuable information not available before.

My only suggestion to improve the scope of the manuscript would be include a paragraph about coinfection dynamics with the other two prominent amphibian pathogens (Bsal and Ranavirus). Even when there is not good information about coinfection between Bd and Bsal/Ranavirus, an attempt to compare the results obtained here with different Bd strains with the existing literature about other coinfection cases would be remarkable.

Finally, my only concern with this work is about how the relative abundance of the three different Bd lineages has been calculated. As the authors say in the text, they determined tadpoles infection status by examining the degree of depigmentation in the oral disc, and genotyped Bd isolates from such tadpoles showing depigmentation. Even when it is true that sometimes Bd infection produces depigmentation, and that Knapp & Morgan proposed depigmentation as a proxy for infection for a particular species, this relationship has not been formally proposed for other species and has not been adopted as a reliable method from other people working with Bd. In fact, Navarro-Lozano (2018; 10.1371/journal.pone.0190955) found no statistical relationship between depigmentation (and other abnormal characteristics of the oral disc) and Bd infection, and specifically in the same study area of Brazil. In addition, and under my experience with Iberian amphibians, even when many Bd infected tadpoles of the *Alytes* genus show depigmentation when reach high Gosner development stages, younger infected tadpoles generally lack this issue. More importantly, depigmentation of the oral disc looks more frequent when *Alytes* tadpoles are infected with the BdGPL strain than when are infected with the less virulent BdCAPE strain. Therefore, if depigmentation of the oral disc varies with the virulence of the Bd strain, the author's data about the relative presence of the 3 different strains would be biased.

Reviewer #1 (Remarks to the Author):

The authors integrate experimental and epidemiological data to provide mechanistic insights into how the within-host competitive interactions could lead to the distribution pattern of Bd lineages across Brazil's Atlantic Forest. They demonstrate that coinfection with chytrid genotypes drives divergent infection dynamics reflecting broad epidemiological patterns.

This is a very interesting study. The study is well performed and the results are clearly described.

This paper will be of interest to a broad audience.

I only have some minor comments:

Material and methods section

Line 173: could the authors provide more details on the inoculum preparation? By scraping the surface, did they collect sporangia? Did they wash the spores before using them?

Thank you for your excellent observation, we did not previously inform the reader if our solutions were filtered or not. Although we gently scraped the surface of the culture plates, we did not detect zoosporangia at the zoospore counting stage performed on the hemocytometer. Therefore, we decided not to wash or filter the inoculums to avoid any possible loss of zoospores. We clarify our inoculum preparation by adding "*We did not filter the zoospore suspensions as zoosporangia were not observed on the hemocytometer.*" to the text (please see line 333-334).

Line 180: did the inocula for the coinfection experiments contain 2.98×10^6 zoospores for each strain or 2.98×10^6 zoospores in total?

Thanks for pointing out this important detail. In coinfection the inoculum dose was 2.98×10^6 zoospores / ml in total. Before performing the exposure experiment, we adjusted all 5 inoculums to the same concentration of 2.98×10^6 zoospores / ml. To perform single infections, we added 1.5 ml of a single inoculum. In coinfections we added 0.75 ml (in double infection) or 0.5 ml (in triple infection) of each inoculum. Thus, the final volume was always 1.5 ml with a concentration of 2.98×10^6 zoospores / ml. However, we understand how confusing this can be, so we added the sentence "(final volume and inoculum dose of 1.5 ml and 2.98×10^6 zoospores / ml, respectively)" in the methods (please see line 340-341).

Reviewer #2 (Remarks to the Author):

The Brazilian Atlantic Forest *Batrachochytrium dendrobatidis* system is one of very few well-documented interlineage contact zones for this globally distributed pathogen. It's wonderful to see more work done on this system, even more so to see this attempt to link up genotype distribution patterns to mechanisms of infection. I am somewhat concerned about components of the study and hope the authors can address my points.

I have reservations regarding the genotyping strategy, how it bears on the detection of recombination in general and how it can be applied to discriminate amongst isolates in some of the experimental treatments. First, can the authors comment on the reliability of the stepwise genotyping strategy in reliably identifying recombinants?

Yes, the SNP assays can reliably distinguish between strains and quantify zoospore equivalents in a mixed sample and they are based on the well-characterized genomes of Bd. The first assay targets a SNP within the mtDNA genome, which does not show recombination in these isolates.

The second targets a SNP in the nuclear genome and can detect and quantify Bd-hybrids (Supplementary Figure S1). Identifying recombinants is, however, beyond the scope of this experiment as it is unlikely that recombination would occur among the isolates we used within our experimental framework and during the timespan of the experiment. We hope our response would successfully answer the reviewer's question.

From my reading, the key assumptions are unidirectional transfer of Bd-Asia-2/Brazil genomes into Bd-GPLs (recombinants only carry GPL mtDNA) and that recombinants can be reliably distinguished from GPLs on the basis of a very limited number of marker loci. I've referred back to O'Hanlon 2018, and the WGS phylogeny indicates that both Brazilian GPLs and Bd-Asia-2/Brazil genomes should carry variation indicative of within-lineage recombination, or the introduction of multiple variants of each lineage into Brazil (eg, Bd-Asia-2/Brazil genomes are not monophyletic within the Bd-Asia-2/Brazil lineage). Either process is likely to include variation that is difficult to identify with simple marker systems. In any case, identifying recombinants using simple marker systems will miss evidence of recombination involving other regions of the genome.

This is a great point. However, since we know the specific isolates that were used in our controlled experiment, we have already identified the recombinant and therefore, do not need to identify the recombinant using this method. This point, however, would likely be an issue in wild-collected amphibians showing evidence of coinfections.

For the experiment, I'm assuming that the missing 80% of frogs in the 3 isolate treatments were heterozygotes for the second step of the qPCR?

We were actually able to determine the outcomes for 80% of the frogs with triple infections; and heterozygotes in the second step of the qPCR would mean the presence of a hybrid isolate (if the initial qPCR only detected homozygous G/G) (Supplementary Figure S1). This nuclear assay allowed us to accurately distinguish between Bd-GPL and hybrid in 100% of the frogs from the double infection and over 80% from the triple infection treatments.

Doesn't the shortcoming of the stepwise genotyping assay mean that it is impossible to clearly discriminate between any mixed GPL and Bd-Asia-2/Brazil infection versus a pure hybrid?

In the wild potentially, but in our experiment we controlled which isolates were used to infect the hosts. The purpose of the stepwise genotyping approach was to detect and also quantify the isolates in a mixed sample based on isolate specific standard curves. In the first qPCR (mtDNA), hybrids will still genotype as G/G. In the second qPCR (nuclear DNA), hybrids will genotype as heterozygotes (A/C) while GPL will genotype as homozygotes (A/A) (Supplementary Figure S1).

In other words, for all of the four 3 isolate treatments, 2 of 6 possible outcomes cannot be discriminated amongst? If this is the case, why was this not discussed in the methods and impacts on analyses clarified?

This is not the case and the purpose of the stepwise genotyping approach was to detect and also quantify the isolates in a mixed sample based on isolate specific standard curves.

It's a red flag for me when your results and discussion involves an argument of competitive suppression of a GPL and in 2 of the 6 treatments involving that isolate you may be underestimating the number of competitive outcomes that unambiguously favor this lineage. It's possible that the reviewer has misinterpreted the genotyping methods and outcomes.

My greatest concern regarding the experiment is that dose is confounded with genotype treatment in the design. I understand why you wanted to make the overall dose strength consistent across treatments, but in so doing you've reduced the concentration of any one isolate by half in the 2 isolate treatments and by 2/3 in the 3 isolate treatments. This means that any effect of treatment on changes in infection outcomes cannot be clearly attributed to competition. The reduction in P1 infectivity in mixed isolate treatments could be driven by the fact these treatments had a reduced dose of P1. There's lots of data out there supporting the relationship between increased virulence and increased dose strength.

These are excellent points and we hope that additional studies on coinfection will address the key question of how to tease apart the effects of dose vs. coinfection *per se*. In designing this study, we reviewed over 20 papers on coinfection from a diverse array of host-pathogen systems, while also keeping theory on the evolutionary epidemiology of virulence at the forefront. We found a mix of experimental designs - with the vast majority using a two-fold increase in pathogen load for coinfection studies. Interestingly, only two or three of these studies mentioned their choice of experimental design, and these authors referenced data demonstrating that the range of pathogen load/dose used in the experiment had a negligible effect on pathogen dynamics, virulence, or final pathogen load.

With Bd, however, there is substantial evidence suggesting that load is strongly and positively correlated with both virulence and final pathogen load. Thus, as the reviewer points out, our study would have been confounded by dose/density-dependence, and potential virulence-transmission trade-offs. We feel that the design of coinfection studies warrants additional attention and we have added additional text to the methods to explicitly state our goals with this particular design and we have also added the following text in the Discussion. Lastly, we have also added an additional paragraph summarizing recent efforts on coinfection in the-amphibian-pathogen systems.

Methods, lines 341-345.

By using such Bd dose we aimed to reduce frog's mortality, while maintaining infection, to increase our ability to track within-hosts pathogen dynamics throughout the experiment. This experimental design enabled us to address the following question. All else equal, how do coinfections influence pathogen growth, transmission potential, and harm to hosts?

Discussion, lines 220-272.

One potential limitation of coinfection experiments, including ours, is that it can be difficult to tease apart the effects of coinfection per se and pathogen load, which is often linked with virulence. We applied the same inoculation dose in both single- and coinfection treatments, which is common in coinfection studies using genotypes of one pathogen species^{23,38,40,41}, though not when competing two pathogen species (e.g.,^{42,43}). This experimental design enabled us to measure competitive interactions between multiple pathogen strains while holding dose — which often positively and strongly correlates with virulence — constant. That is, we asked, all else equal, how does coinfection influence pathogen growth, transmission potential, and virulence?

An alternative question that is also addressed in coinfection studies is how does a competitive advantage/disadvantage (based on initial dose) influence competitive interaction, transmission potential, and virulence. Many of these studies manipulate coinfection with a two-fold increase in pathogen dose. In this case, coinfection is still confounded by potential density-dependent effects or potential dose-virulence-transmission interactions. These concerns can be limited by a careful selection of experimental doses such that a two-fold increase in the experimental dose has a negligible effect on pathogen dynamics, virulence, or final pathogen load^{6,36}. For the Bd system, however, an increase in initial pathogen load is positively correlated with increased virulence and pathogen load⁴⁴. Hence, we chose the more conservative experimental design to hold total pathogen load constant.

Viewed in this light, neither study design can tease apart the effect of dose on coinfection and both designs are hampered by the possibility that density-dependent effects could drive outcomes either by increasing harm to hosts through a dose response or by suppressing/releasing pathogen growth due to density-dependence. An important gap in knowledge is whether these two experimental designs reveal important differences from transient dynamics early in infection versus equilibrium levels after an infection has established⁴⁵. Nonetheless, both designs address relevant and interesting questions. However, addressing each of these questions in a single study is logistically challenging, especially when considering the sample sizes required to address this question with the number of pathogen genotypes examined here.

Moreover, while neither of these conditions likely mirror infections outside of the laboratory, they are experimentally tractable and commonly used (see Supplementary Table 1); ideally, one would examine co-infections in a response-surface design with the pathogen

serovars or strains manipulated across a wide-range of doses. Unfortunately, for most in vivo systems, including the focal system here, the latter design is logistically challenging and rarely examined. In reviewing over 20 co-infection studies, only one study manipulated dose across single- and mixed-infections⁴⁶. Our study, however, takes an important step in addressing one component of the coinfection puzzle and highlights important areas for future research.

*Also, amphibians are recognized as an ideal system to assess the ecological and evolutionary implications of coinfections to both the infectious agent and the host⁴⁷, as they are commonly infected by multiple parasites (e.g., ⁴⁸⁻⁵¹) and have experienced disease-induced population declines globally¹⁴. Though, elucidating the factors shaping within-host interactions and scales (spatial and/or biological) at which coinfections influence disease dynamics in amphibians has proven to be highly complex and inconsistent, which has hampered our ability to identify general trends thus far (reviewed in ⁴⁷). Previous work assessing coinfections by amphibian parasites associated with population declines (e.g., *Bd*, *Bsal* and *Ranavirus*) have determined that density-dependent competition among parasites⁵², order of parasite arrival^{53,54} and timing of exposure between parasites^{53,55} play key roles in parasite colonization and transmission potential, and influence amphibians' response and survival. Hence, our study adds to our current understanding by providing evidence of within-host competitive and facilitative interactions between genotypes of *Bd*, which can influence disease dynamics, virulence evolution and shape the genetic structure of the pathogen population.*

The link between the experiment and the spatial data set is not really investigated. There have been a few studies of the spatial distribution of *Bd* genotypes in Brazil, but it's unclear to me how the experimental design is set up to address spatial patterns. As far as I am aware,

arguments have been made regarding environmental envelopes and host specificity. For example, Greenspan et al (2018) showed how comparative patterns of virulence were reversed between GPL and recombinants in different native host species. The title of the paper includes ‘reflecting broad epidemiological patterns’, but I don’t see these patterns investigated at all here. We agree with the reviewer's comment and have added additional details and updated the text. In the Introduction, we remind the reader that the panzootic lineage may have replaced the enzootic lineage throughout the Atlantic Forest, and that coinfections were likely an important factor in the epidemiological outcome. In the discussion, we highlight how our field data agree with our experimental results, both suggesting the competitive advantage of the panzootic lineage and the replacement of the enzootic lineage, which perhaps still resists through facilitative interactions. In the methods, we added a sentence acknowledging the potential limitations of our data, and that we conducted a preliminary attempt to link the field data with the experimental results. Finally, we changed our title to: "*Coinfection with chytrid genotypes drives divergent infection dynamics reflecting regional distribution patterns*".

Introduction (lines 70-73)

This observation suggests that coinfections may either lead to more severe infections or select for more virulent lineages, which can lead to the replacement of less competitive enzootic genotypes in the pathogen population.

Discussion (lines 171-177)

Importantly, the outcome of the within-host competitive interactions observed in our study, and in Jenkinson et al. ¹⁷, may be captured by the observed distribution of Bd lineages, where Bd-

GPL showed a much broader distribution compared to the enzootic lineage and the hybrid (Fig. 1B). Combined, these results suggest the likely scenario that upon arrival Bd-GPL rapidly spread throughout the Atlantic Forest outcompeting the enzootic lineage^{10,17}, which may be persisting up to this day, in part, due to within-host facilitative interactions.

Methods (lines 432-433)

We used the genotyping results from this survey as a preliminary attempt to link the results from the exposure experiment to field data.

The study boils down to a single experiment, and in that experiment, possible outcomes of 4 of 14 treatments cannot be identified, and the fundamental premise that outcomes are driven by competition cannot be fully tested due to confounding effects. I would be really happy if the authors can change my perceptions and look forward to seeing the revisions that will do so.

Reviewer #3 (Remarks to the Author):

The work of Carvalho et al. investigates how coinfections with 3 different Bd strains (BdGPL, BdBrazil and its hybrid) shaped virulence and transmission. They used both experimental and epidemiological data to elucidate the implications for disease dynamics and virulence evolution. The manuscript is really well written and the obtained results are interesting and certainly deserve publication. The performed experiment is well planned and executed, and it really brings valuable information not available before. My only suggestion to improve the scope of the manuscript would be include a paragraph about coinfection dynamics with the other two prominent amphibian pathogens (Bsal and Ranavirus). Even when there is not good information

about coinfection between Bd and Bsal/Ranavirus, an attempt to compare the results obtained here with different Bd strains with the existing literature about other coinfection cases would be remarkable.

We agree with the reviewer's suggestion about adding a paragraph about coinfection dynamics in other important amphibian parasites. Thus, we have included a paragraph in the Discussion section that highlights, in a concise way, some of the key factors that are considered to shape the outcome of coinfection between amphibian parasites (see below; and in the lines 258-272 of the revised manuscript).

Discussion (lines 258-272):

Also, amphibians are recognized as an ideal system to assess the ecological and evolutionary implications of coinfections to both the infectious agent and the host⁴⁷, as they are commonly infected by multiple parasites (e.g.,⁴⁸⁻⁵¹) and have experienced disease-induced population declines globally¹⁴. Though, elucidating the factors shaping within-host interactions and scales (spatial and/or biological) at which coinfections influence disease dynamics in amphibians has proven to be highly complex and inconsistent, which has hampered our ability to identify general trends thus far (reviewed in⁴⁷). Previous work assessing coinfections by amphibian parasites associated with population declines (e.g., Bd, Bsal and Ranavirus) have determined that density-dependent competition among parasites⁵², order of parasite arrival^{53,54} and timing of exposure between parasites^{53,55} play key roles in parasite colonization and transmission potential, and influence amphibians' response and survival. Hence, our study adds to our current understanding by providing evidence of within-host competitive and facilitative

interactions between genotypes of Bd, which can influence disease dynamics, virulence evolution and shape the genetic structure of the pathogen population.

Finally, my only concern with this work is about how the relative abundance of the three different Bd lineages has been calculated. As the authors say in the text, they determined tadpoles infection status by examining the degree of depigmentation in the oral disc, and genotyped Bd isolates from such tadpoles showing depigmentation. Even when it is true that sometimes Bd infection produces depigmentation, and that Knapp & Morgan proposed depigmentation as a proxy for infection for a particular species, this relationship has not been formally proposed for other species and has not been adopted as a reliable method from other people working with Bd. In fact, Navarro-Lozano (2018; 10.1371/journal.pone.0190955) found no statistical relationship between depigmentation (and other abnormal characteristics of the oral disc) and Bd infection, and specifically in the same study area of Brazil. In addition, and under my experience with Iberian amphibians, even when many Bd infected tadpoles of the *Alytes* genus show depigmentation when reach high Gosner development stages, younger infected tadpoles generally lack this issue. More importantly, depigmentation of the oral disc looks more frequent when *Alytes* tadpoles are infected with the BdGPL strain than when are infected with the less virulent BdCAPE strain. Therefore, if depigmentation of the oral disc varies with the virulence of the Bd strain, the author's data about the relative presence of the 3 different strains would be biased.

Jaime Bosch

In terms of the reviewer's concern about the potential limitation of the technique we used to diagnose the infection status of tadpoles, we agree that this is important and worthy of

recognition in the manuscript. Hence, we have added several lines in the method section (see below and in the lines 430-437) that acknowledged such potential limitations. In addition, we also highlighted in this section that this is a first step to link the experiment result with field data. Thus, we hope we have addressed the reviewer observation.

Methods (lines 430-437)

We determined tadpole infection status by examining the degree of dekeratinization in the oral disc⁷², and used this approach to select tadpoles for Bd isolation. We used the genotyping results from this survey as a preliminary attempt to link the results from the exposure experiment to field data. However, our ability to qualitatively make such a link may be limited by the potential variation in depigmentation rates across Bd strains/lineages; and also, by the inconsistent relationship between infection status and degree of dekeratinization of tadpole mouthparts in some species from the Atlantic Forest⁷⁷.

Reviewers' comments:

Reviewer #2 (Remarks to the Author):

I was really happy to see the explanation of the diagnostics for the isolates in competitive assays, I'm comfortable with that. However, the explanation for why dose could be ignored isn't satisfactory. It's hard to 'see' the comparative effects in Fig 2-3 as the labelling isn't clear, but the fact remains that to generate mixed inoculates isolates had to be diluted by a factor of 2 or 3. There's no reason to assume that the shape of dose response curves across isolates is constant, rather there is published evidence that the slopes of Bd growth rates vary both across lineages and within the GPL clade. If growth functions vary across the experimental isolates, then dose response curves will also have different slopes, with the result that at different dose strengths, competitive outcomes will be largely determined by dose strength rather than competitive ability.

Given the experimental design, this is impossible to disentangle. Author statements that dose was held constant are invalid, what they really mean is the overall concentrations of zoospores was kept constant, but in reality the dose strength for any one isolate was varied significantly. I feel they have not dealt with this issue, and as a result do not recommend acceptance due to inappropriate experimental design.

Reviewer #3 (Remarks to the Author):

The authors have satisfactorily addressed my concerns and I am happy with the revised version of the manuscript.

Congratulations to the authors for this relevant work.

Reviewer #1 (Remarks to the Author):

The authors integrate experimental and epidemiological data to provide mechanistic insights into how the within-host competitive interactions could lead to the distribution pattern of Bd lineages across Brazil's Atlantic Forest. They demonstrate that coinfection with chytrid genotypes drives divergent infection dynamics reflecting broad epidemiological patterns.

This is a very interesting study. The study is well performed and the results are clearly described.

This paper will be of interest to a broad audience.

I only have some minor comments:

Material and methods section

Line 173: could the authors provide more details on the inoculum preparation? By scraping the surface, did they collect sporangia? Did they wash the spores before using them?

Thank you for your excellent observation, we did not previously inform the reader if our solutions were filtered or not. Although we gently scraped the surface of the culture plates, we did not detect zoosporangia at the zoospore counting stage performed on the hemocytometer. Therefore, we decided not to wash or filter the inoculums to avoid any possible loss of zoospores. We clarify our inoculum preparation by adding "*We did not filter the zoospore suspensions as zoosporangia were not observed on the hemocytometer.*" to the text (please see line 333-334).

Line 180: did the inocula for the coinfection experiments contain 2.98×10^6 zoospores for each strain or 2.98×10^6 zoospores in total?

Thanks for pointing out this important detail. In coinfection the inoculum dose was 2.98×10^6 zoospores / ml in total. Before performing the exposure experiment, we adjusted all 5 inoculums to the same concentration of 2.98×10^6 zoospores / ml. To perform single infections, we added 1.5 ml of a single inoculum. In coinfections we added 0.75 ml (in double infection) or 0.5 ml (in triple infection) of each inoculum. Thus, the final volume was always 1.5 ml with a concentration of 2.98×10^6 zoospores / ml. However, we understand how confusing this can be, so we added the sentence "(final volume and inoculum dose of 1.5 ml and 2.98×10^6 zoospores / ml, respectively)" in the methods (please see line 340-341).

Reviewer #2 (Remarks to the Author):

The Brazilian Atlantic Forest *Batrachochytrium dendrobatidis* system is one of very few well-documented interlineage contact zones for this globally distributed pathogen. It's wonderful to see more work done on this system, even more so to see this attempt to link up genotype distribution patterns to mechanisms of infection. I am somewhat concerned about components of the study and hope the authors can address my points.

I have reservations regarding the genotyping strategy, how it bears on the detection of recombination in general and how it can be applied to discriminate amongst isolates in some of the experimental treatments. First, can the authors comment on the reliability of the stepwise genotyping strategy in reliably identifying recombinants?

Yes, the SNP assays can reliably distinguish between strains and quantify zoospore equivalents in a mixed sample and they are based on the well-characterized genomes of Bd. The first assay targets a SNP within the mtDNA genome, which does not show recombination in these isolates.

The second targets a SNP in the nuclear genome and can detect and quantify Bd-hybrids (Supplementary Figure S1). Identifying recombinants is, however, beyond the scope of this experiment as it is unlikely that recombination would occur among the isolates we used within our experimental framework and during the timespan of the experiment. We hope our response would successfully answer the reviewer's question.

From my reading, the key assumptions are unidirectional transfer of Bd-Asia-2/Brazil genomes into Bd-GPLs (recombinants only carry GPL mtDNA) and that recombinants can be reliably distinguished from GPLs on the basis of a very limited number of marker loci. I've referred back to O'Hanlon 2018, and the WGS phylogeny indicates that both Brazilian GPLs and Bd-Asia-2/Brazil genomes should carry variation indicative of within-lineage recombination, or the introduction of multiple variants of each lineage into Brazil (eg, Bd-Asia-2/Brazil genomes are not monophyletic within the Bd-Asia-2/Brazil lineage). Either process is likely to include variation that is difficult to identify with simple marker systems. In any case, identifying recombinants using simple marker systems will miss evidence of recombination involving other regions of the genome.

This is a great point. However, since we know the specific isolates that were used in our controlled experiment, we have already identified the recombinant and therefore, do not need to identify the recombinant using this method. This point, however, would likely be an issue in wild-collected amphibians showing evidence of coinfections.

For the experiment, I'm assuming that the missing 80% of frogs in the 3 isolate treatments were heterozygotes for the second step of the qPCR?

We were actually able to determine the outcomes for 80% of the frogs with triple infections; and heterozygotes in the second step of the qPCR would mean the presence of a hybrid isolate (if the initial qPCR only detected homozygous G/G) (Supplementary Figure S1). This nuclear assay allowed us to accurately distinguish between Bd-GPL and hybrid in 100% of the frogs from the double infection and over 80% from the triple infection treatments.

Doesn't the shortcoming of the stepwise genotyping assay mean that it is impossible to clearly discriminate between any mixed GPL and Bd-Asia-2/Brazil infection versus a pure hybrid?

In the wild potentially, but in our experiment we controlled which isolates were used to infect the hosts. The purpose of the stepwise genotyping approach was to detect and also quantify the isolates in a mixed sample based on isolate specific standard curves. In the first qPCR (mtDNA), hybrids will still genotype as G/G. In the second qPCR (nuclear DNA), hybrids will genotype as heterozygotes (A/C) while GPL will genotype as homozygotes (A/A) (Supplementary Figure S1).

In other words, for all of the four 3 isolate treatments, 2 of 6 possible outcomes cannot be discriminated amongst? If this is the case, why was this not discussed in the methods and impacts on analyses clarified?

This is not the case and the purpose of the stepwise genotyping approach was to detect and also quantify the isolates in a mixed sample based on isolate specific standard curves.

It's a red flag for me when your results and discussion involves an argument of competitive suppression of a GPL and in 2 of the 6 treatments involving that isolate you may be underestimating the number of competitive outcomes that unambiguously favor this lineage. It's possible that the reviewer has misinterpreted the genotyping methods and outcomes.

My greatest concern regarding the experiment is that dose is confounded with genotype treatment in the design. I understand why you wanted to make the overall dose strength consistent across treatments, but in so doing you've reduced the concentration of any one isolate by half in the 2 isolate treatments and by 2/3 in the 3 isolate treatments. This means that any effect of treatment on changes in infection outcomes cannot be clearly attributed to competition. The reduction in P1 infectivity in mixed isolate treatments could be driven by the fact these treatments had a reduced dose of P1. There's lots of data out there supporting the relationship between increased virulence and increased dose strength.

These are excellent points and we hope that additional studies on coinfection will address the key question of how to tease apart the effects of dose vs. coinfection *per se*. In designing this study, we reviewed over 20 papers on coinfection from a diverse array of host-pathogen systems, while also keeping theory on the evolutionary epidemiology of virulence at the forefront. We found a mix of experimental designs - with the vast majority using a two-fold increase in pathogen load for coinfection studies. Interestingly, only two or three of these studies mentioned their choice of experimental design, and these authors referenced data demonstrating that the range of pathogen load/dose used in the experiment had a negligible effect on pathogen dynamics, virulence, or final pathogen load.

With Bd, however, there is substantial evidence suggesting that load is strongly and positively correlated with both virulence and final pathogen load. Thus, as the reviewer points out, our study would have been confounded by dose/density-dependence, and potential virulence-transmission trade-offs. We feel that the design of coinfection studies warrants additional attention and we have added additional text to the methods to explicitly state our goals with this particular design and we have also added the following text in the Discussion. Lastly, we have also added an additional paragraph summarizing recent efforts on coinfection in the-amphibian-pathogen systems.

Methods, lines 341-345.

By using such Bd dose we aimed to reduce frog's mortality, while maintaining infection, to increase our ability to track within-hosts pathogen dynamics throughout the experiment. This experimental design enabled us to address the following question. All else equal, how do coinfections influence pathogen growth, transmission potential, and harm to hosts?

Discussion, lines 220-272.

One potential limitation of coinfection experiments, including ours, is that it can be difficult to tease apart the effects of coinfection per se and pathogen load, which is often linked with virulence. We applied the same inoculation dose in both single- and coinfection treatments, which is common in coinfection studies using genotypes of one pathogen species^{23,38,40,41}, though not when competing two pathogen species (e.g., ^{42,43}). This experimental design enabled us to measure competitive interactions between multiple pathogen strains while holding dose — which often positively and strongly correlates with virulence — constant. That is, we asked, all else equal, how does coinfection influence pathogen growth, transmission potential, and virulence?

An alternative question that is also addressed in coinfection studies is how does a competitive advantage/disadvantage (based on initial dose) influence competitive interaction, transmission potential, and virulence. Many of these studies manipulate coinfection with a two-fold increase in pathogen dose. In this case, coinfection is still confounded by potential density-dependent effects or potential dose-virulence-transmission interactions. These concerns can be limited by a careful selection of experimental doses such that a two-fold increase in the experimental dose has a negligible effect on pathogen dynamics, virulence, or final pathogen load^{6,36}. For the Bd system, however, an increase in initial pathogen load is positively correlated with increased virulence and pathogen load⁴⁴. Hence, we chose the more conservative experimental design to hold total pathogen load constant.

Viewed in this light, neither study design can tease apart the effect of dose on coinfection and both designs are hampered by the possibility that density-dependent effects could drive outcomes either by increasing harm to hosts through a dose response or by suppressing/releasing pathogen growth due to density-dependence. An important gap in knowledge is whether these two experimental designs reveal important differences from transient dynamics early in infection versus equilibrium levels after an infection has established⁴⁵. Nonetheless, both designs address relevant and interesting questions. However, addressing each of these questions in a single study is logistically challenging, especially when considering the sample sizes required to address this question with the number of pathogen genotypes examined here.

Moreover, while neither of these conditions likely mirror infections outside of the laboratory, they are experimentally tractable and commonly used (see Supplementary Table 1); ideally, one would examine co-infections in a response-surface design with the pathogen

serovars or strains manipulated across a wide-range of doses. Unfortunately, for most in vivo systems, including the focal system here, the latter design is logistically challenging and rarely examined. In reviewing over 20 co-infection studies, only one study manipulated dose across single- and mixed-infections⁴⁶. Our study, however, takes an important step in addressing one component of the coinfection puzzle and highlights important areas for future research.

*Also, amphibians are recognized as an ideal system to assess the ecological and evolutionary implications of coinfections to both the infectious agent and the host⁴⁷, as they are commonly infected by multiple parasites (e.g., ⁴⁸⁻⁵¹) and have experienced disease-induced population declines globally¹⁴. Though, elucidating the factors shaping within-host interactions and scales (spatial and/or biological) at which coinfections influence disease dynamics in amphibians has proven to be highly complex and inconsistent, which has hampered our ability to identify general trends thus far (reviewed in ⁴⁷). Previous work assessing coinfections by amphibian parasites associated with population declines (e.g., *Bd*, *Bsal* and *Ranavirus*) have determined that density-dependent competition among parasites⁵², order of parasite arrival^{53,54} and timing of exposure between parasites^{53,55} play key roles in parasite colonization and transmission potential, and influence amphibians' response and survival. Hence, our study adds to our current understanding by providing evidence of within-host competitive and facilitative interactions between genotypes of *Bd*, which can influence disease dynamics, virulence evolution and shape the genetic structure of the pathogen population.*

The link between the experiment and the spatial data set is not really investigated. There have been a few studies of the spatial distribution of *Bd* genotypes in Brazil, but it's unclear to me how the experimental design is set up to address spatial patterns. As far as I am aware,

arguments have been made regarding environmental envelopes and host specificity. For example, Greenspan et al (2018) showed how comparative patterns of virulence were reversed between GPL and recombinants in different native host species. The title of the paper includes ‘reflecting broad epidemiological patterns’, but I don’t see these patterns investigated at all here. We agree with the reviewer's comment and have added additional details and updated the text. In the Introduction, we remind the reader that the panzootic lineage may have replaced the enzootic lineage throughout the Atlantic Forest, and that coinfections were likely an important factor in the epidemiological outcome. In the discussion, we highlight how our field data agree with our experimental results, both suggesting the competitive advantage of the panzootic lineage and the replacement of the enzootic lineage, which perhaps still resists through facilitative interactions. In the methods, we added a sentence acknowledging the potential limitations of our data, and that we conducted a preliminary attempt to link the field data with the experimental results. Finally, we changed our title to: "*Coinfection with chytrid genotypes drives divergent infection dynamics reflecting regional distribution patterns*".

Introduction (lines 70-73)

This observation suggests that coinfections may either lead to more severe infections or select for more virulent lineages, which can lead to the replacement of less competitive enzootic genotypes in the pathogen population.

Discussion (lines 171-177)

Importantly, the outcome of the within-host competitive interactions observed in our study, and in Jenkinson et al. ¹⁷, may be captured by the observed distribution of Bd lineages, where Bd-

GPL showed a much broader distribution compared to the enzootic lineage and the hybrid (Fig. 1B). Combined, these results suggest the likely scenario that upon arrival Bd-GPL rapidly spread throughout the Atlantic Forest outcompeting the enzootic lineage^{10,17}, which may be persisting up to this day, in part, due to within-host facilitative interactions.

Methods (lines 432-433)

We used the genotyping results from this survey as a preliminary attempt to link the results from the exposure experiment to field data.

The study boils down to a single experiment, and in that experiment, possible outcomes of 4 of 14 treatments cannot be identified, and the fundamental premise that outcomes are driven by competition cannot be fully tested due to confounding effects. I would be really happy if the authors can change my perceptions and look forward to seeing the revisions that will do so.

Reviewer #3 (Remarks to the Author):

The work of Carvalho et al. investigates how coinfections with 3 different Bd strains (BdGPL, BdBrazil and its hybrid) shaped virulence and transmission. They used both experimental and epidemiological data to elucidate the implications for disease dynamics and virulence evolution. The manuscript is really well written and the obtained results are interesting and certainly deserve publication. The performed experiment is well planned and executed, and it really brings valuable information not available before. My only suggestion to improve the scope of the manuscript would be include a paragraph about coinfection dynamics with the other two prominent amphibian pathogens (Bsal and Ranavirus). Even when there is not good information

about coinfection between Bd and Bsal/Ranavirus, an attempt to compare the results obtained here with different Bd strains with the existing literature about other coinfection cases would be remarkable.

We agree with the reviewer's suggestion about adding a paragraph about coinfection dynamics in other important amphibian parasites. Thus, we have included a paragraph in the Discussion section that highlights, in a concise way, some of the key factors that are considered to shape the outcome of coinfection between amphibian parasites (see below; and in the lines 258-272 of the revised manuscript).

Discussion (lines 258-272):

Also, amphibians are recognized as an ideal system to assess the ecological and evolutionary implications of coinfections to both the infectious agent and the host⁴⁷, as they are commonly infected by multiple parasites (e.g.,⁴⁸⁻⁵¹) and have experienced disease-induced population declines globally¹⁴. Though, elucidating the factors shaping within-host interactions and scales (spatial and/or biological) at which coinfections influence disease dynamics in amphibians has proven to be highly complex and inconsistent, which has hampered our ability to identify general trends thus far (reviewed in⁴⁷). Previous work assessing coinfections by amphibian parasites associated with population declines (e.g., Bd, Bsal and Ranavirus) have determined that density-dependent competition among parasites⁵², order of parasite arrival^{53,54} and timing of exposure between parasites^{53,55} play key roles in parasite colonization and transmission potential, and influence amphibians' response and survival. Hence, our study adds to our current understanding by providing evidence of within-host competitive and facilitative

interactions between genotypes of Bd, which can influence disease dynamics, virulence evolution and shape the genetic structure of the pathogen population.

Finally, my only concern with this work is about how the relative abundance of the three different Bd lineages has been calculated. As the authors say in the text, they determined tadpoles infection status by examining the degree of depigmentation in the oral disc, and genotyped Bd isolates from such tadpoles showing depigmentation. Even when it is true that sometimes Bd infection produces depigmentation, and that Knapp & Morgan proposed depigmentation as a proxy for infection for a particular species, this relationship has not been formally proposed for other species and has not been adopted as a reliable method from other people working with Bd. In fact, Navarro-Lozano (2018; 10.1371/journal.pone.0190955) found no statistical relationship between depigmentation (and other abnormal characteristics of the oral disc) and Bd infection, and specifically in the same study area of Brazil. In addition, and under my experience with Iberian amphibians, even when many Bd infected tadpoles of the *Alytes* genus show depigmentation when reach high Gosner development stages, younger infected tadpoles generally lack this issue. More importantly, depigmentation of the oral disc looks more frequent when *Alytes* tadpoles are infected with the BdGPL strain than when are infected with the less virulent BdCAPE strain. Therefore, if depigmentation of the oral disc varies with the virulence of the Bd strain, the author's data about the relative presence of the 3 different strains would be biased.

Jaime Bosch

In terms of the reviewer's concern about the potential limitation of the technique we used to diagnose the infection status of tadpoles, we agree that this is important and worthy of

recognition in the manuscript. Hence, we have added several lines in the method section (see below and in the lines 430-437) that acknowledged such potential limitations. In addition, we also highlighted in this section that this is a first step to link the experiment result with field data. Thus, we hope we have addressed the reviewer observation.

Methods (lines 430-437)

We determined tadpole infection status by examining the degree of dekeratinization in the oral disc⁷², and used this approach to select tadpoles for Bd isolation. We used the genotyping results from this survey as a preliminary attempt to link the results from the exposure experiment to field data. However, our ability to qualitatively make such a link may be limited by the potential variation in depigmentation rates across Bd strains/lineages; and also, by the inconsistent relationship between infection status and degree of dekeratinization of tadpole mouthparts in some species from the Atlantic Forest⁷⁷.

Reviewers' comments:

Reviewer #4 (Remarks to the Author):

In this manuscript, the authors manipulate coinfection of different chytrid strains and observe the outcome of infection on virulence and strain specific loads. This is a great example of applying coinfection ideas to a topic and system has real-world consequences for amphibian survival in the wild. As a late reviewer to the manuscript, I have tried to restrict my comments to the coinfection experimental design and interpretation.

The key experimental design issue is that the initial pathogen dose is kept constant, and so any outcome of coinfection is the product of varying starting loads (1/2 for P1 and P2 in coinfection) as well as any possible competitive interactions. This is an approach that is not uncommon in coinfection studies of this type, largely due to the fact that the sample size doubles (here would be 30+ treatments) to do the other required treatment combination, where total load changes as well. But more some studies with this design have more extensively justified why this may not be an issue for their system, such as providing evidence that starting densities did not matter for endpoint doses (e.g., Gibson Evolution 2019)

Without this extra information, it is more difficult to conclude that the performance of P1 declined in coinfection. If the starting dose does matters, then the lower growth and virulence of P1 might emerge naturally without the competitor strain having to do anything. Perhaps the authors have data or can point to another study that shows this sort of variation in starting load is not relevant in the end (such as from a dose-response curve study, like some cited in table 1 of ref#44). Even the load of P1 and P2 at half does would really help with this.

At the very least, I think this could be addressed more clearly (and earlier) in the discussion. Rather than discussing why such an approach (of also varying starting dose) was not taken (and for good reasons, I understand why it was not), I think more focus could be placed on first, what can't be ruled out in explaining the results. Then hopefully providing a better justification for why it might not matter in the end. The current argument that starting dose is correlated with reduced survival is less clear than suggested (which is actually a good thing), at least based on my read of ref#44 (i.e. figure 1, where the x and y axes should be the other way around, has a lot of overlap in what starting doses lead to survival costs and which do not. Also, what about the fact that P1 in double and triple infection is pretty similar (which presumably has a lower starting dose as well).

My only other main comment is about the interpretation of virulence in a coinfection needs some clarification. Throughout the manuscript when referring to the virulence of the coinfection (as opposed to the virulence of the pathogen in isolation) it might be worth using "overall virulence" to help make this distinction clear. e.g., Line 35 and 83 etc. Statements like "The presence of multiple genotypes reduced the virulence (i.e., increased host life-span and survival) of the most virulent genotype (P1)" (Line 686)" are not necessarily true. What changed was the overall expression of virulence in the host (this need to be made clear), and this is decoupled from what might happen to a specific strain and virulence evolution as a result (as discussed in ref #5). It all depends on the underlying mechanism of confection, which cannot really be determined here. Unless I am missing something key about how the virulence of just P1 could be isolated out from a coinfection of P1 + competitors?

Other remarks regarding readability:

- The first paragraph gives the impression coinfection will always select for more virulent pathogens. It would be useful to make it explicitly that this is the case when competing over host resources (see arguments in ref 5 etc). The introduction is generally missing some nuanced discussion that the overall virulence from coinfections can be decoupled from the virulence of any invading pathogen that is observed when it is isolated. Overall virulence can go up, down, or be intermediate to the original

virulence depending on which mechanism of the interaction is at play (see discussion in Alizon 2013 or Bashey 2015 etc). This helps set up some of the results.

- The axes labels for the figures are virulence, but the actual trait is lifespan. Either use a common control lifespan to plot the reduction (or plot death rate, $1/\text{lifespan}$) so that higher values equals higher virulence, or stick with host lifespan only label. Otherwise, it is misleading and confusing, as I expected higher y-values to be more virulent.

- One other suggestion for the figures is to make Figure 2 all about pathogen load (final and competitiveness), which can be attributed directly to P1 or P2 in each case. And then have virulence and survival as figure 3 (or vice versa), which can only be assessed at the host level (i.e. overall outcomes). The x-axis label in Figure 2 "P1 in single, double, and triple infections" applies to Bd load but not virulence (on my read at least, see above). This could be made clearer.

Comments from Reviewer 4

1. In this manuscript, the authors manipulate coinfection of different chytrid strains and observe the outcome of infection on virulence and strain specific loads. This is a great example of applying coinfection ideas to a topic and system has real-world consequences for amphibian survival in the wild. As a late reviewer to the manuscript, I have tried to restrict my comments to the coinfection experimental design and interpretation.

The key experimental design issue is that the initial pathogen dose is kept constant, and so any outcome of coinfection is the product of varying starting loads (1/2 for P1 and P2 in coinfection) as well as any possible competitive interactions. This is an approach that is not uncommon in coinfection studies of this type, largely due to the fact that the sample size doubles (here would be 30+ treatments) to do the other required treatment combination, where total load changes as well. But more some studies with this design have more extensively justified why this may not be an issue for their system, such as providing evidence that starting densities did not matter for endpoint doses (e.g., Gibson Evolution 2019).

Without this extra information, it is more difficult to conclude that the performance of P1 declined in coinfection. If the starting dose does matters, then the lower growth and virulence of P1 might emerge naturally without the competitor strain having to do anything. Perhaps the authors have data or can point to another study that shows this sort of variation in starting load is not relevant in the end (such as from a dose-response curve study, like some cited in table 1 of ref#44). Even the load of P1 and P2 at half does would really help with this.

At the very least, I think this could be addressed more clearly (and earlier) in the discussion. Rather than discussing why such an approach (of also varying starting dose) was not taken (and for good reasons, I understand why it was not), I think more focus could be placed on first, what can't be ruled out in explaining the results. Then hopefully providing a better justification for why it might not matter in the end. The current argument that starting dose is correlated with reduced survival is less clear than suggested (which is actually a good thing), at least based on my read of ref#44 (i.e. figure 1, where the x and y axes should be the other way around, has a lot of overlap in what starting doses lead to survival costs and which do not. Also, what about the fact that P1 in double and triple infection is pretty similar (which presumably has a lower starting dose as well).

We thank the reviewer for highlighting this important point. We have substantially revised the Manuscript to address these key points.

(i) We have now included an additional statistical analyses and figures (please see first section of Supplementary Material and Supplementary Figure 1A-C) that explicitly examined the effect of initial zoospore dose on host life span (a metric of overall virulence), competitiveness (early establishment), and transmission potential (total pathogen load). We have also reported the results from this analysis early in the Results section (please see lines 94 – 101).

(ii): In addition to the additional analysis, we have kept, and improved, a thorough explanation of this key logical constraint associated with our experimental design in the Discussion section (lines 236 – 275), and added a justification in the Methods section (lines 361 – 365).

2. Also, what about the fact that P1 in double and triple infection is pretty similar (which presumably has a lower starting dose as well).

Great point! Our substantial revisions throughout the text and the additional figures and statistical analyses were designed, in large part, to address this important point. We feel that these revisions helped to clarify that overall, any changes in single versus coinfection treatments was found to depend on the underlying pathogen genotype. The initial pathogen dose contributed little to host life span, competitiveness, or transmission potential.

Also, and as a comment, the interpretation of this result supports the idea of one of the main findings of this study, which is that we showed opposite patterns in the infection dynamics of mixed-infection between P1 and P2 (e.g., the observed pattern in transmission potential, Figure 2E and F), despite having the same starting dose of the reference genotypes.

3. My only other main comment is about the interpretation of virulence in a coinfection needs some clarification. Throughout the manuscript when referring to the virulence of the coinfection (as opposed to the virulence of the pathogen in isolation) it might be worth using “overall virulence” to help make this distinction clear. e.g., Line 35 and 83 etc. Statements like “The presence of multiple genotypes reduced the virulence (i.e., increased host life-span and survival) of the most virulent genotype (P1)” (Line 686)” are not necessarily true. What changed was the overall expression of virulence in the host (this need to be made clear), and this is decoupled from what might happen to a specific strain and virulence evolution as a result (as discussed in ref #5). It all depends on the underlying mechanism of coinfection, which cannot really be determined here. Unless I am missing something key about how the virulence of just P1 could be isolated out from a coinfection of P1 + competitors?

Thank you for your excellent observation. We have modified all text to include "overall virulence" when referring to the virulence of coinfections/pathogens and have carefully corrected sentences such as the one exemplified by the reviewer. For example, “*The presence of multiple genotypes reduced the overall virulence (i.e., increased host life-span and survival) in those treatments involving the most virulent genotype (P1; B and E).*”

4. The first paragraph gives the impression coinfection will always select for more virulent pathogens. It would be useful to make it explicitly that this is the case when competing over host resources (see arguments in ref 5 etc). The introduction is generally missing some nuanced discussion that the overall virulence from coinfections can be decoupled from the virulence of any invading pathogen that is observed when it is isolated. Overall virulence can go up, down, or be intermediate to the original virulence depending on which mechanism of the interaction is at play (see discussion in Alizon 2013 or Bashey 2015 etc). This helps set up some of the results.

Thanks for pointing out this important detail. We have modified the first paragraph of the Introduction to cover the various outcomes of arising from coinfection (Lines: 54-57). We have

also clarified that we are referring to competition for resources by adding "resources" to line 49.

5. The axes labels for the figures are virulence, but the actual trait is lifespan. Either use a common control lifespan to plot the reduction (or plot death rate, $1/\text{lifespan}$) so that higher values equals higher virulence, or stick with host lifespan only label. Otherwise, it is misleading and confusing, as I expected higher y-values to be more virulent.

Agreed. We modified the legend to "Host lifespan".

6. One other suggestion for the figures is to make Figure 2 all about pathogen load (final and competitiveness), which can be attributed directly to P1 or P2 in each case. And then have virulence and survival as figure 3 (or vice versa), which can only be assessed at the host level (i.e. overall outcomes). The x-axis label in Figure 2 "P1 in single, double, and triple infections" applies to Bd load but not virulence (on my read at least, see above). This could be made clearer.

We agree again and thank the reviewer for this constructive and helpful feedback. Figures 2 and 3 have been updated to show pathogen load (Figure 2) and host survival (Figure 3).

REVIEWERS' COMMENTS:

Reviewer #4 (Remarks to the Author):

First, I must apologise for the delay in getting my review back – getting COVID finally, and then having it go through the family, made for a difficult few weeks. I thank the authors for their careful response and willingness to make changes, it is much appreciated. I think the dose issue is now addressed in a very positive way and the new analysis and figures clear. I just have one last very minor request, and that is to make it clear-cut that higher virulence = lower lifespan at every opportunity.

For example, it might be worth adding in explicitly that the most virulent pathogen is the one with the lowest lifespan. e.g., added to Line 112: The most competitive genotype, the panzootic P1, was also the most virulent one (i.e. the lowest average lifespan) whereas the enzootic and hybrid genotypes were less competitive than the panzootic genotypes and showed intermediate virulence. And later, Line 127, "P1 virulence was higher than overall virulence in both mean lifespan" is slight confusing. "P1 virulence was higher than overall virulence leading to a greater reduction in lifespan" or something of the like. There may be a few other instances of this sort of thing as well that will help reinforce that the lowest lifespan is the greatest virulence (even though this is a logical conclusion anyway).

Comments from Reviewer 4

Reviewer #4 (Remarks to the Author):

First, I must apologise for the delay in getting my review back – getting COVID finally, and then having it go through the family, made for a difficult few weeks. I thank the authors for their careful response and willingness to make changes, it is much appreciated. I think the dose issue is now addressed in a very positive way and the new analysis and figures clear. I just have one last very minor request, and that is to make it clear-cut that higher virulence = lower lifespan at every opportunity.

For example, it might be worth adding in explicitly that the most virulent pathogen is the one with the lowest lifespan. e.g., added to Line 112: The most competitive genotype, the panzootic P1, was also the most virulent one (i.e. the lowest average lifespan) whereas the enzootic and hybrid genotypes were less competitive than the panzootic genotypes and showed intermediate virulence. And later, Line 127, "P1 virulence was higher than overall virulence in both mean lifespan" is slight confusing. "P1 virulence was higher than overall virulence leading to a greater reduction in lifespan" or something of the like. There may be a few other instances of this sort of thing as well that will help reinforce that the lowest lifespan is the greatest virulence (even though this is a logical conclusion anyway).

We thank the reviewer for taking the time to review our manuscript and we are glad to hear that the reviewer has recovered from COVID-19. We have now added the reviewer suggestion (i.e. to highlight the link between virulence and reduced host life span) to the sentences specified in the comment above (please see lines 112-113, and line 128 of the revised version). Also, we revised the manuscript to look for other parts of the text in which we could highlight the link between virulence and reduced life span, however, we think that by highlighting this link in the parts suggested by the reviewer (early in the result section) lays out this relationship in the most appropriate section of the manuscript.